# Implicit regularization via Spectral Neural Networks and non-linear matrix sensing

## Abstract

The phenomenon of *implicit regularization* has attracted interest in recent years as a fundamental aspect of the remarkable generalizing ability of neural networks. In a nutshell, it entails that gradient flow dynamics in many neural nets, even without any explicit regularizer in the loss function, converges to the solution of a regularized learning problem. However, known results attempting to theoretically explain this phenomenon focus overwhelmingly on the setting of linear neural nets, and the simplicity of the linear structure is particularly crucial to existing arguments. In this paper, we explore this problem in the context of more realistic neural networks with a general class of non-linear activation functions, and rigorously demonstrate the implicit regularization phenomenon for such networks in the setting of matrix sensing problems. This is coupled with rigorous rate guarantees that ensure exponentially fast convergence of gradient descent, complemented by matching lower bounds which stipulate that the exponential rate is the best achievable. In this vein, we contribute a network architecture called Spectral Neural Networks (*abbrv.* SNN) that is particularly suitable for matrix learning problems. Conceptually, this entails coordinatizing the space of matrices by their singular values and singular vectors, as opposed to by their entries, a potentially fruitful perspective for matrix learning. We demonstrate that the SNN architecture is inherently much more amenable to theoretical analysis than vanilla neural nets and confirm its effectiveness in the context of matrix sensing, supported via both mathematical guarantees and empirical investigations. We believe that the SNN architecture has the potential to be of wide applicability in a broad class of matrix learning scenarios.

## 1 Introduction

A longstanding pursuit of deep learning theory is to explain the astonishing ability of neural networks to generalize despite having far more learnable parameters than training data, even in the absence of any explicit regularization. An established understanding of this phenomenon is that the gradient descent algorithm induces a so-called *implicit regularization* effect. In very general terms, implicit regularization entails that gradient flow dynamics in many neural nets, even without any explicit regularizer in the loss function, converges to the solution of a regularized learning problem. In a sense, this creates a learning paradigm that automatically favors models characterized by "low complexity".

A standard test-bed for mathematical analysis in studying implicit regularization in deep learning is the *matrix sensing* problem. The goal is to approximate a matrix $X^\star$ from a set of measurement matrices $A_1, \ldots, A_m$ and observations $y_1, \ldots, y_m$ where $y_i = \langle A_i, X^\star \rangle$. A common approach, *matrix factorization*, parameterizes the solution as a product matrix, i.e., $X = UV^\top$, and optimizes the resulting non-convex objective to fit the data. This is equivalent to training a depth-2 neural network with a linear activation function.

In an attempt to explain the generalizing ability of over-parameterized neural networks, Neyshabur et al. (2014) first suggested the idea of an implicit regularization effect of the optimizer, which entails a bias towards solutions that generalize well. Gunasekar et al. (2017) investigated the possibility of an implicit norm-regularization effect of gradient descent in the context of shallow matrix factorization. In particular, they studied the standard Burer-Monteiro approach Burer & Monteiro (2003) to matrix factorization, which may be viewed as a depth-2 linear neural network. They were able to

theoretically demonstrate an implicit norm-regularization phenomenon, where a suitably initialized gradient flow dynamics approaches a solution to the nuclear-norm minimization approach to low-rank matrix recovery Recht et al. (2010), in a setting where the involved measurement matrices commute with each other. They also conjectured that this latter restriction on the measurement matrices is unnecessary. This conjecture was later resolved by Li et al. (2018) in the setting where the measurement matrices satisfy a restricted isometry property. Other aspects of implicit regularization in matrix factorization problems were investigated in several follow-up papers (Neyshabur et al., 2017; Arora et al., 2019; Razin & Cohen, 2020; Tarmoun et al., 2021; Razin et al., 2021). For instance, Arora et al. (2019) showed that the implicit norm-regularization property of gradient flow, as studied by Gunasekar et al. (2017), does not hold in the context of deep matrix factorization. Razin & Cohen (2020) constructed a simple $2 \times 2$ example, where the gradient flow dynamics lead to an eventual blow-up of any matrix norm, while a certain relaxation of rank—the so-called *e-rank*—is minimized in the limit. These works suggest that implicit regularization in deep networks should be interpreted through the lens of rank minimization, not norm minimization. Incidentally, Razin et al. (2021) have recently demonstrated similar phenomena in the context of tensor factorization.

Researchers have also studied implicit regularization in several other learning problems, including linear models (Soudry et al., 2018; Zhao et al., 2019; Du & Hu, 2019), neural networks with one or two hidden layers (Li et al., 2018; Blanc et al., 2020; Gidel et al., 2019; Kubo et al., 2019; Saxe et al., 2019). Besides norm-regularization, several of these works demonstrate the implicit regularization effect of gradient descent in terms of other relevant quantities such as margin (Soudry et al., 2018), the number of times the model changes its convexity (Blanc et al., 2020), linear interpolation (Kubo et al., 2019), or structural bias (Gidel et al., 2019).

A natural use case scenario for investigating the implicit regularization phenomenon is the problem of matrix sensing. Classical works in matrix sensing and matrix factorization utilize convex relaxation approaches, i.e., minimizing the nuclear norm subject to agreement with the observations, and deriving tight sample complexity bound (Srebro & Shraibman, 2005; Candès & Recht, 2009; Recht et al., 2010; Candès & Tao, 2010; Keshavan et al., 2010; Recht, 2011). Recently, many works analyze gradient descent on the matrix sensing problem. Ge et al. (2016) and Bhojanapalli et al. (2016) show that the non-convex objectives on matrix sensing and matrix completion with low-rank parameterization do not have any spurious local minima. Consequently, the gradient descent algorithm converges to the global minimum.

Despite the large body of works studying implicit regularization, most of them consider the linear setting. It remains an open question to understand the behavior of gradient descent in the presence of non-linearities, which are more realistic representations of neural nets employed in practice.

In this paper, we make an initial foray into this problem, and investigate the implicit regularization phenomenon in more realistic neural networks with a general class of non-linear activation functions. We rigorously demonstrate the occurrence of an implicit regularization phenomenon in this setting for matrix sensing problems, reinforced with quantitative rate guarantees ensuring exponentially fast convergence of gradient descent to the best approximation in a suitable class of matrices. Our convergence upper bounds are complemented by matching lower bounds which demonstrate the optimality of the exponential rate of convergence.

In the bigger picture, we contribute a network architecture that we refer to as the Spectral Neural Network architecture (abbrv. SNN), which is particularly suitable for matrix learning scenarios. Conceptually, this entails coordinatizing the space of matrices by their singular values and singular vectors, as opposed to by their entries. We believe that this point of view can be beneficial for tackling matrix learning problems in a neural network setup. SNNs are particularly well-suited for theoretical analysis due to the spectral nature of their non-linearities, as opposed to vanilla neural nets, while at the same time provably guaranteeing effectiveness in matrix learning problems. We also introduce a much more general counterpart of the near-zero initialization that is popular in related literature, and our methods are able to handle a much more robust class of initializing setups that are constrained only via certain inequalities. Our theoretical contributions include a compact analytical representation of the gradient flow dynamics, accorded by the spectral nature of our network architecture. We demonstrate the efficacy of the SNN architecture through its application to the matrix sensing problem, bolstered via both theoretical guarantees and extensive empirical studies. We believe that the SNN architecture has the potential to be of wide applicability in a broad class of matrix learning problems. In particular, we believe that the SNN architecture would be natural for the study of rank (or e-rank)

minimization effect of implicit regularization in deep matrix/tensor factorization problems, especially given the fact that e-rank is essentially a spectrally defined concept.

## 2 PROBLEM SETUP

Let $X^\star \in \mathbb{R}^{d_1 \times d_2}$ be an unknown rectangular matrix that we aim to recover. Let $A_1, \ldots, A_m \in \mathbb{R}^{d_1 \times d_2}$ be $m$ measurement matrices, and the label vector $y \in \mathbb{R}^m$ is generated by

$$y_i = \langle A_i, X^\star \rangle, \tag{1}$$

where $\langle A, B \rangle = \operatorname{tr}(A^\top B)$ denotes the Frobenius inner product. We consider the following squared loss objective

$$\ell(X) := \frac{1}{2} \sum_{i=1}^m (y_i - \langle A_i, X \rangle)^2. \tag{2}$$

This setting covers problems including matrix completion (where the $A_i$'s are indicator matrices), matrix sensing from linear measurements, and multi-task learning (in which the columns of $X$ are predictors for the tasks, and $A_i$ has only one non-zero column). We are interested in the regime where $m \ll d_1 \times d_2$, i.e., the number of measurements is much less than the number of entries in $X^\star$, in which case 2 is under-determined with many global minima. Therefore, merely minimizing 2 does not guarantee correct recovery or good generalization.

Following previous works, instead of working with $X$ directly, we consider a non-linear factorization of $X$ as follows

$$X = \sum_{k=1}^K \alpha_k \Gamma(U_k V_k^\top), \tag{3}$$

where $\alpha \in \mathbb{R}$, $U_k \in \mathbb{R}^{d_1 \times d}$, $V_k \in \mathbb{R}^{d_2 \times d}$, and the matrix-valued function $\Gamma : \mathbb{R}^{d_1 \times d_2} \to \mathbb{R}^{d_1 \times d_2}$ transforms a matrix by applying a nonlinear real-valued function $\gamma(\cdot)$ on its singular values. We focus on the over-parameterized setting $d \geq d_2 \geq d_1$, i.e., the factorization does not impose any rank constraints on $X$. Let $\alpha = \{\alpha_1, \ldots, \alpha_K\}$ be the collection of the $\alpha_k$'s. Similarly, we define $U$ and $V$ to be the collections of $U_k$'s and $V_k$'s.

### 2.1 GRADIENT FLOW

For each $k \in [K]$, let $\alpha_k(t), U_k(t), V_k(t)$ denote the trajectories of gradient flow, where $\alpha_k(0), U_k(0), V_k(0)$ are the initial conditions. Consequently, $X(t) = \sum_{k=1}^K \alpha_k(t) \Gamma(U_k(t) V_k(t)^\top)$. The dynamics of gradient flow is given by the following differential equations, for $k \in [K]$

$$\partial_t \alpha_k = -\nabla_{\alpha_k} \ell(X(t)), \quad \partial_t U_k = -\nabla_{U_k} \ell(X(t)), \quad \partial_t V_k = -\nabla_{V_k} \ell(X(t)). \tag{4}$$

## 3 THE SNN ARCHITECTURE

In this work, we contribute a novel neural network architecture, called the Spectral Neural Network (*abbrv.* SNN), that is particularly suitable for matrix learning problems. At the fundamental level, the SNN architecture entails an application of a non-linear activation function on a matrix-valued input in the spectral domain. This may be followed by a linear combination of several such spectrally transformed matrix-structured data.

To be precise, let us focus on an elemental neuron, which manipulates a single matrix-valued input $X$. If we have a singular value decomposition $X = \Phi \bar{X} \Psi^\top$, where $\Phi, \Psi$ are orthogonal matrices and $\bar{X}$ is the diagonal matrix of singular values of $X$. Let $\gamma$ be any activation function of choice. Then the elemental neuron acts on $X$ as follows :

$$X \mapsto \Phi \, \gamma(\bar{X}) \, \Psi^\top, \tag{5}$$

where $\gamma(\bar{X})$ is a diagonal matrix with the non-linearity $\gamma$ applied entrywise to the diagonal of $\bar{X}$.

A *block* in the SNN architecture comprises of $K \geq 1$ elemental neurons as above, taking in $K$ matrix-valued inputs $X_1, \ldots, X_K$. Each input matrix $X_i$ is then individually operated upon by an elemental

**Anatomy of a block**

**Spectral Neural Network Architecture**

Figure 1: Visualization of the anatomy of an SNN block and a depth-$D$ SNN architecture. Each SNN block takes as input $K$ matrices and outputs one matrix, both the input and output matrices are of size $\mathbb{R}^{d_1 \times d_2}$. In layer $i$ of the SNN, there are $L_i$ blocks, which aggregate matrices from the previous layers to produce $L_i$ output matrices as inputs for the next layer. The number of input matrices to a block equals the number of neurons in the previous layer. For example, blocks in layer 1 have $K = L_0$, blocks in layer 2 have $K = L_1$, and blocks in layer $i$ have $K = L_{i-1}$.

neuron, and finally, the resulting matrices are aggregated linearly to produce a matrix-valued output for the block. The coefficients of this linear combination are also parameters in the SNN architecture, and are to be learned during the process of training the network.

The comprehensive SNN architecture is finally obtained by combining such blocks into multiple layers of a deep network, as illustrated in 1.

## 4  MAIN RESULTS

For the purposes of theoretical analysis, in the present article, we specialize the SNN architecture to focus on the setting of (quasi-) commuting measurement matrices $A_i$ and spectral near zero initialization; c.f. Assumptions 1 and 2 below. Similar settings have attracted considerable attention in the literature, including the foundational works of Gunasekar et al. (2017) and Arora et al. (2019). Furthermore, our analysis holds under very general analytical requirements on the activation function $\gamma$; see Assumption 3 in the following.

**Assumption 1.** The measurement matrices $A_1, \ldots, A_m$ share the same left and right singular vectors. Specifically, there exists two orthogonal matrices $\Phi \in \mathbb{R}^{d_1 \times d_1}$ and $\Psi \in \mathbb{R}^{d_2 \times d_2}$, and a sequence of (rectangular) diagonal matrices [1] $\bar{A}_1, \ldots, \bar{A}_m \in \mathbb{R}^{d_1 \times d_2}$ such that for any $i \in [m]$, we have

$$A_i = \Phi \bar{A}_i \Psi^\top. \tag{6}$$

Let $\sigma^{(i)}$ be the vector containing the singular values of $A_i$, i.e., $\sigma^{(i)} = \text{diag}(\bar{A}_i)$. Furthermore, we assume that there exist real coefficients $a_1, \ldots, a_m$ that

$$a_1 \sigma^{(1)} + \cdots + a_m \sigma^{(m)} = \mathbb{1}. \tag{7}$$

---

[1] Rectangular diagonal matrices arise in the singular value decomposition of rectangular matrices, see Appendix D.

We let $\overline{X}^\star = \Phi^\top X^\star \Psi$ and $\sigma^\star$ be the vector containing the diagonal elements of $\overline{X}^\star$, i.e., $\sigma^\star = \mathrm{diag}(\overline{X}^\star)$. Without loss of generality, we may also assume that $\sigma^\star$ is coordinatewise non-zero. This can be easily ensured by adding the rectangular identity matrix (c.f. Appendix D) $cI_{d_1 \times d_2}$ to $X^*$ for some large enough positive number $c$.

Eq. 6 postulates that the measurement matrices share the same (left- and right-) singular vectors. This holds if and only if the measurement matrices pair-wise *quasi-commute* in the sense that for any $i, j \in [m]$, we have

$$A_i A_j^\top = A_j A_i^\top, \quad A_i^\top A_j = A_j^\top A_i. \tag{8}$$

A natural class of examples of such quasi-commuting measurement matrices comes from families of commuting projections. In such a scenario Eq. 7 stipulates that these projections cover all the coordinate directions, which may be related conceptually to a notion of the measurements being sufficiently informative. For example, in this setting, Eq. 7 would entail that the trace of $X^\star$ can be directly computed on the basis of the measurements.

Eq. 7 acts as a regularity condition on the singular values of the measurement matrices. For example, it prohibits peculiar scenarios where $\sigma_1^{(i)} = 0$ for all $i$, i.e., all measurement matrices have 0 as their smallest singular values, which makes it impossible to sense the smallest singular value of $X^\star$ from linear measurements.

Note that

$$y_i = \langle A_i, X^\star \rangle = \mathrm{Tr}(A_i^\top X^\star) = \mathrm{Tr}(\Psi \bar{A}_i \Phi^\top X^\star) = \mathrm{Tr}(\bar{A}_i \Phi^\top X^\star \Psi) = \langle \bar{A}_i, \Phi^\top X^\star \Psi \rangle, \tag{9}$$

where in the above we use the fact that $\bar{A}_i = \bar{A}_i^\top$ (since $\bar{A}_i$ is diagonal) and the cyclic property of trace. We have

$$y_i = \langle \bar{A}_i, \overline{X}^\star \rangle = \langle \sigma^{(i)}, \sigma^\star \rangle = \sigma^{(i)\top} \sigma^\star, \tag{10}$$

where the second equality is due to $\bar{A}_i$ being diagonal.

We further define vectors $z^{(k)}$ and three matrices $B, \mathcal{C}$, and $H$ as follows

$$z_i^{(k)} = [\bar{U}_k]_{ii} [\bar{V}_k]_{ii}$$
$$B = \left[ \sigma^{(1)} \mid \ldots \mid \sigma^{(m)} \right] \in \mathbb{R}^{d_1 \times m}$$
$$\mathcal{C} = BB^\top \in \mathbb{R}^{d_1 \times d_1}$$
$$H = \left[ \gamma(z^{(1)}) \mid \ldots \mid \gamma(z^{(K)}) \right] \in \mathbb{R}^{d_1 \times K}.$$

Under these new notations, we can write the label vector $y$ as $y = B^\top \sigma^\star$.

**Assumption 2. (Spectral Initialization)** Let $\Phi$ and $\Psi$ be the matrices containing the left and right singular vectors of the measurement matrices from Assumption 1. Let $G \in \mathbb{R}^{d \times d}$ is any arbitrary orthogonal matrix. We initialize $X(0)$ such that the following condition holds: for any $k = 1, \ldots, K$, we have

(a) $U_k(0) = \Phi \bar{U}_k(0) G$ and $V_k = \Psi \bar{V}(0) G$, and

(b) $\bar{U}_k(0)$ and $\bar{V}_k(0)$ are diagonal, and

(c) $\sum_{k=1}^K \alpha_k \gamma(\bar{U}_k(0)_{ii} \bar{V}_k(0)_{ii}) \leq \sigma_i^\star$ for any $i = 1, \ldots, d_1$.

Assumption 2, especially part (c) therein, may be thought of as a much more generalized counterpart of the "near-zero" initialization which is widely used in the related literature (Gunasekar et al. (2017); Li et al. (2018); Arora et al. (2019)). A direct consequence of Assumption 2 is that at initialization, the matrix $X(0)$ has $\Phi$ and $\Psi$ as its left and right singular vectors. As we will see later, this initialization imposes a distinctive structure on the gradient flow dynamics, allowing for an explicit analytical expression for the flow of each component of $X$.

**Assumption 3.** The function $\gamma : \mathbb{R} \to \mathbb{R}$ is bounded between $[0, 1]$, and is differentiable and non-decreasing on $\mathbb{R}$.

Assumption 3 imposes regularity conditions on the non-linearity $\gamma$. Common non-linearities that are used in deep learning such as Sigmoid, ReLU or tanh satisfy the differentiability and non-decreasing conditions, while the boundedness can be achieved by truncating the outputs of these functions if necessary.

Our first result provides a compact representation of the gradient flow dynamics in suitable coordinates. The derivation of this dynamics involves matrix differentials utilizing the Khatri-Rao product $\Psi \boxtimes \Phi$ of the matrices $\Psi$ and $\Phi$ (see Eq. 21 in Appendix A of the supplement).

**Theorem 1.** *Suppose Assumptions 1, 2, 3 hold. Then the gradient flow dynamics in 4 are*

$$\partial_t \boldsymbol{\alpha} = H^\top \mathcal{C}(\sigma^\star - H\alpha),$$
$$\partial_t U_k = \Phi L_k \Psi^\top V_k, \text{ and}$$
$$\partial_t V_k = (\Phi L_k \Psi^\top)^\top U_k,$$

*where $L_k \in \mathbb{R}^{d_1 \times d_2}$ is a diagonal matrix whose diagonal is given by*

$$\text{diag}(L_k) = \lambda^{(k)} = [\lambda_1^{(k)}, \ldots, \lambda_{d_1}^{(k)}]^\top = \alpha_k \gamma'(z^{(k)}) \circ \mathcal{C}(\sigma^\star - H\alpha). \tag{11}$$

*Proof.* (Main ideas – full details in Appendix A). We leverage the fact that the non-linearity $\gamma(\cdot)$ only changes the singular values of the product matrix $U_k V_k^\top$ while keeping the singular vectors intact. Therefore, the gradient flow in 4 preserves the left and right singular vectors. Furthermore, by Assumption 2, $U_k V_k^\top$ has $\Phi$ and $\Psi$ as its left and right singular vectors at initialization, which remains the same throughout. This property also percolates to $\nabla_{U_k V_k^\top} \ell$. Mathematically speaking, $\nabla_{U_k V_k^\top} \ell$ becomes diagonalizable by $\Phi$ and $\Psi$, i.e.,

$$\nabla_{U_k V_k^\top} \ell = \Phi \Lambda_k \Psi^\top$$

for some diagonal matrix $\Lambda_k$. It turns out that $\Lambda_k = L_k$ as given in the statement of the theorem. In view of Eq. 4, this explains the expressions for $\partial_t U_k$ and $\partial_t V_k$. Finally, since $\alpha_k$ is a scalar, the partial derivative of $\ell$ with respect to $\alpha_k$ is relatively straightforward to compute. $\square$

Theorem 1 provides closed-form expressions for the dynamics of the individual components of $X$, namely $\alpha_k, U_k$ and $V_k$. We want to highlight that the compact analytical expression and the simplicity of the gradient flow dynamics on the components are a direct result of the spectral non-linearity. In other words, if we use the conventional element-wise non-linearity commonly used in deep learning, the above dynamics will be substantially more complicated, containing several Hadamard products and becoming prohibitively harder for theoretical analysis.

As a direct corollary of Theorem 1, the gradient flow dynamics on $\bar{U}_k$ and $\bar{V}_k$ are

$$\partial_t \bar{U}_k = L_k \bar{V}_k, \quad \partial_t \bar{V}_k = L_k^\top \bar{U}_k. \tag{12}$$

Under Assumption 2, $\bar{U}_k(0)$ and $\bar{V}_k(0)$ are diagonal matrices. From the gradient flow dynamics in Eq. 12, and recalling that the $L_k$'s are diagonal, we infer that $\partial_t \bar{U}_k(0)$ and $\partial_t \bar{V}_k(0)$ are also diagonal. Consequently, $\bar{U}_k(t)$ and $\bar{V}_k(t)$ remain diagonal for all $t \geq 0$ since the gradient flow dynamics in Eq. 12 does not induce any change in the off-diagonal elements. Thus, $\bar{U}_k(t)\bar{V}(t)_k^\top$ also remains diagonal throughout.

A consequence of the spectral initialization is that the left and right singular vectors of $X(t)$ stay constant at $\Phi$ and $\Psi$ throughout the entire gradient flow procedure. To this end, the gradient flow dynamics is completely determined by the evolution of the singular values of $X(t)$, i.e., $H\alpha$. The next result characterizes the convergence of the singular values of $X(t)$.

**Theorem 2.** *Under Assumptions 1 and 2, for any $i = 1, \ldots, d_1$, there are constants $\eta_i, C_i > 0$ such that we have :*

$$0 \leq \sigma_i^\star - (H(t)\alpha(t))_i \leq C_i e^{-\eta_i t}.$$

*On the other hand, we have the lower bound*

$$\|\sigma^\star - (H(t)\alpha(t))\|_2 \geq C e^{-\eta t},$$

*for some constants $\eta, C > 0$.*

*Proof.* (Main ideas – full details in Appendix B). By part (c) of Assumption 2, at initialization, we have that $H(0)\alpha(0) \leq_e \sigma^\star$, in which the symbol $\leq_e$ denotes the element-wise less than or equal to relation. Therefore, to prove Theorem 2, it is sufficient to show that $H(t)\alpha(t)$ is increasing to $\sigma^\star$ element-wise at an exponential rate. To achieve that, we show that the evolution of $H\alpha$ over time can be expressed as

$$\partial_t(H\alpha) = 4 \sum_{k=1}^{K} \alpha_k^2 \cdot \left( \gamma'(z^{(k)})^2 \circ \mathcal{C}(\sigma^\star - H\alpha) \circ \Big( \int \lambda^{(k)} \circ z^{(k)} dt + C^{(k)} \Big) \right) + H H^\top \mathcal{C}(\sigma^\star - H\alpha).$$

By definition, the matrix $B$ contains the singular values of the $A_i$'s, and therefore its entries are non-negative. Consequently, since $\mathcal{C} = BB^\top$, the entries of $\mathcal{C}$ are also non-negative. By Assumption 3, we have that $\gamma(\cdot) \in [0, 1]$, thus the entries of $H$ are non-negative. Finally, by Assumption 2, we have $H(0)\alpha(0) < \sigma^\star$ entry-wise. For these reasons, each entry in $\partial_t(H\alpha)$ is non-negative at initialization, and indeed, for each $i$, the quantity $(H\alpha)_i$ is increasing as long as $(H\alpha)_i < \sigma_i^*$. As $H\alpha$ approaches $\sigma^\star$, the gradient $\partial_t(H\alpha)$ decreases. If it so happened that $H\alpha = \sigma^\star$ at some finite time, then $\partial_t(H\alpha)$ would exactly equal 0, which would then cause $H\alpha$ to stay constant at $\sigma^*$ from then on.

Thus, each $(H\alpha)_i$ is non-decreasing and bounded above by $\sigma_i^*$, and therefore must converge to a limit $\leq \sigma_i^*$. If this limit was strictly smaller than $\sigma_i^*$, then by the above argument $(H\alpha)_i$ would be still increasing, indicating that this cannot be the case. Consequently, we may deduce that

$$\lim_{t \to \infty} H(t)\alpha(t) = \sigma^\star.$$

It remains to show that the convergence is exponentially fast. To achieve this, we show in the detailed proof that each entry of $\partial_t H\alpha$ is not only non-negative but also bounded away from 0, i.e.,

$$\partial_t(H\alpha)_i \geq \eta_i(\sigma_i^\star - (H\alpha)_i),$$

for some constant $\eta_i > 0$. This would imply that $H\alpha$ converges to $\sigma^\star$ at an exponential rate. $\square$

The limiting matrix output by the network is, therefore, $\Phi \mathrm{Diag}(\sigma^*)\Psi^\top$, and given the fact that $\sigma^* = \mathrm{diag}(\Phi^\top X^*\Psi)$, this would be the best approximation of $X^*$ among matrices with (the columns of) $\Phi$ and $\Psi$ as their left and right singular vectors. This is perhaps reasonable, given the fact that the sensing matrices $A_i$ also share the same singular vectors, and it is natural to expect an approximation that is limited by their properties. In particular, when the $A_i$ are symmetric and hence commuting, under mild linear independence assumptions, $\Phi \mathrm{Diag}(\sigma^*)\Psi^\top$ would be the best approximation of $X^*$ in the algebra generated by the $A_i$-s, which is again a natural class given the nature of the measurements.

We are now ready to rigorously demonstrate the phenomenon of implicit regularization in our setting. To this end, following the gradient flow dynamics, we are interested in the behavior of $X_\infty$ in the limit when time goes to infinity.

**Theorem 3.** *Let $X_\infty = \lim_{t \to \infty} X(t)$. Under Assumptions 1 and 2, the following hold:*

(a) *$\ell(X_\infty) = 0$, and*

(b) *$X_\infty$ solves the optimization problem*

$$\min_{X \in \mathbb{R}^{d_1 \times d_2}} \|X\|_* \quad \text{subject to} \quad y_i = \langle A_i, X \rangle \ \forall i \in [m]. \tag{13}$$

*Proof.* (Main ideas – full details in Appendix C). A direct corollary of Theorem 2 is that

$$H(\infty)\alpha(\infty) = \sigma^\star.$$

By some algebraic manipulations, we can show that the limit of $X$ takes the form

$$X_\infty = \Phi \Big[ \mathrm{Diag}(\sigma^\star) \Big] \Psi^\top.$$

Now, let us look at the prediction given by $X_\infty$. For any $i = 1, \ldots, m$, we have

$$\langle A_i, X_\infty \rangle = \langle \Phi \bar{A}_i \Psi^\top, \Phi \mathrm{Diag}(\sigma^\star)\Psi^\top \rangle = \langle \sigma^{(i)}, \sigma^\star \rangle = y_i,$$

where the last equality holds due to 10. This implies that $\ell(X_\infty) = 0$, proving (a).

To prove (b), we will show that $X_\infty$ satisfies the Karush-Kuhn-Tucker (KKT) conditions of the optimization problem stated in Eq. 74. The conditions are

$$\exists \nu \in \mathbb{R}^m \text{ s.t.} \quad \forall i \in [m], \langle A_i, X \rangle = y_i \quad \text{and} \quad \nabla_X \|X\|_* + \sum_{i=1}^m \nu_i A_i = 0.$$

The solution matrix $X_\infty$ satisfies the first condition as proved in part (a). As for the second condition, note first that the gradient of the nuclear norm of $X_\infty$ is given by

$$\nabla \|X_\infty\| = \Phi \Psi^\top.$$

Therefore the second condition becomes

$$\Phi \Psi^\top + \sum_{i=1}^m \nu_i A_i = 0 \quad \Leftrightarrow \quad \Phi \Big( I - \sum_{i=1}^m \nu_i \bar{A}_i \Big) \Psi^\top = 0 \quad \Leftrightarrow \quad B\nu = \mathbb{1}.$$

However, by Assumption 1, the vector $\mathbb{1}$ lies in the column space of $B$, which implies the existence of such a vector $\nu$. This concludes the proof of part (b). $\qquad\square$

## 5 NUMERICAL STUDIES

In this section, we present numerical studies to complement our theoretical analysis. Additional experiments on the multi-layer SNN architecture, as well as with relaxed assumptions, can be found in Appendix E.

We highlight that gradient flow can be viewed as gradient descent with an infinitesimal learning rate. Therefore, the gradient flow model only acts as a good proxy to study gradient descent when the learning rate is sufficiently small. Throughout our experiments, we shall consider gradient descent with varying learning rates, and demonstrate that the behavior suggested by our theory is best achieved using small learning rates.

We generate the true matrix by sampling each entry of $X^\star$ independently from a standard Gaussian distribution, suitably normalized. For every measurement matrix $A_i$, $i = 1, \ldots, m$, we sample each entry of the diagonal matrix $\bar{A}_i$ from the uniform distribution on $(0, 1)$, sort them in decreasing order, and set $A_i = \Phi \bar{A}_i \Psi^\top$, where $\Phi$ and $\Psi$ are randomly generated orthogonal matrices. We then record the measurements $y_i = \langle A_i, X^\star \rangle$, $i = 1, \ldots, m$. For every $k = 1, \ldots, K$, we initialize $\bar{U}(0)$ and $\bar{V}(0)$ to be diagonal matrices, whose diagonal entries are sampled uniformly from $(0, 10^{-3})$, and sorted in descending order. Similarly, $\alpha_k$ is also sampled uniformly from $(0, 10^{-3})$. We take $d_1 = d_2 = 10$ (thus $X^\star$ has 100 entries), $m = 60$ measurement matrices, and $K = 3$. As for the non-linearity, we use the sigmoid $\gamma(x) = 1/(1 + e^{-x})$.

In the first experiment (c.f. Fig. 2), we empirically demonstrate that the singular values of the solution matrices converge to $\sigma^\star$ at an exponential rate as suggested by Theorem 2. From the leftmost plot of Fig. 2, we observe that when running gradient descent with a small learning rate, i.e., $10^{-4}$, the singular values of $X$ converges to $\sigma^\star$ exponentially fast. By visual inspection, it takes only less than 4000 iterations of gradient descent for the singular values of $X$ to converge. As we increase the learning rate, the convergence rate slows down significantly, as demonstrated by the middle and rightmost plots of Fig. 2. For the learning rates of $10^{-3}$ and $10^{-2}$, it takes approximately 6000 and more than 10000 iterations respectively to converge. We re-emphasize that our theoretical results are for gradient flow, which only acts as a good surrogate to study gradient descent when the learning rates are infinitesimally small. As a result, our theory cannot characterize the behavior of gradient descent algorithm with substantially large learning rates.

In the second experiment (c.f. Fig. 3), we show the evolution of the nuclear norm over time. Interestingly, but perhaps not surprisingly, the choice of the learning rate dictates the speed of convergence. Moderate values of the learning rate seem to yield the quickest convergence to stationarity.

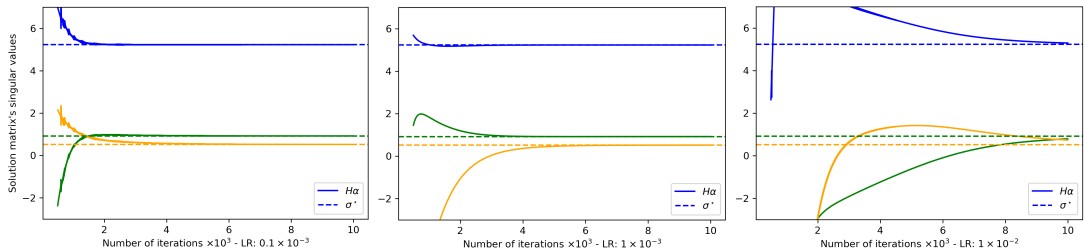

Figure 2: The evolution of $X$ largest 3 singular values when running gradient descent on the matrix sensing problem with learning rates of different magnitudes. The first 100 iterations have vastly different values and are omitted for clarity of presentation.

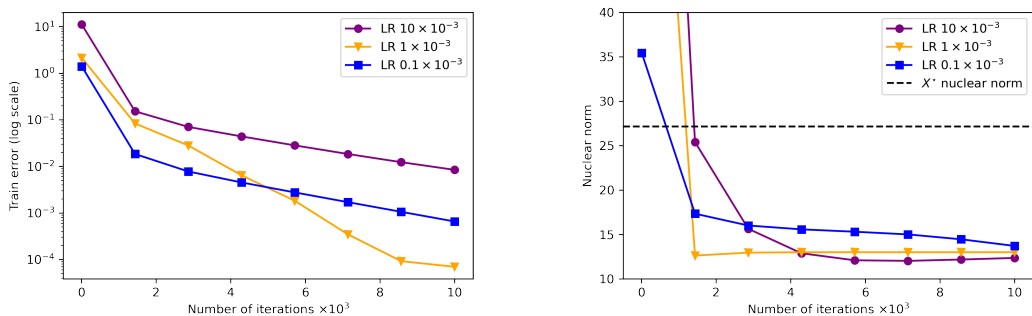

Figure 3: The evolution of $X$'s training error (left panel) and nuclear norm (right panel) over time with learning rates (LR) of different magnitudes.

## 6   SUMMARY OF CONTRIBUTIONS AND FUTURE DIRECTIONS

In this work, we investigate the phenomenon of implicit regularization via gradient flow in neural networks, using the problem of matrix sensing as a canonical test bed. We undertake our investigations in the more realistic scenario of non-linear activation functions, compared to the mostly linear structure that has been explored in the literature. In this endeavor, we contribute a novel neural network architecture called Spectral Neural Network (SNN) that is particularly well-suited for matrix learning problems. SNNs are characterized by a spectral application of a non-linear activation function to matrix-valued input, rather than an entrywise one. Conceptually, this entails coordinatizing the space of matrices by their singular values and singular vectors, as opposed to by their entries. We believe that this perspective has the potential to gain increasing salience in a wide array of matrix learning scenarios. SNNs are particularly well-suited for theoretical analysis due to their spectral nature of the non-linearities, as opposed to vanilla neural nets, while at the same time provably guaranteeing effectiveness in matrix learning problems. We also introduce a much more general counterpart of the near-zero initialization that is popular in related literature, and our methods are able to handle a much more robust class of initializing setups that are constrained only via certain inequalities. Our theoretical contributions include a compact analytical representation of the gradient flow dynamics, accorded by the spectral nature of our network architecture. We demonstrate a rigorous proof of exponentially fast convergence of gradient descent to an approximation to the original matrix that is best in a certain class, complemented by a matching lower bound, Finally, we demonstrate the matrix-valued limit of the gradient flow dynamics achieves zero training loss and is a minimizer of the matrix nuclear norm, thereby rigorously establishing the phenomenon of implicit regularization in this setting.

Our work raises several exciting possibilities for follow-up and future research. A natural direction is to extend our analysis to extend our detailed analysis to the most general setting when the sensing matrices $A_i$ are non-commuting. An investigation of the dynamics in the setting of discrete time gradient descent (as opposed to continuous time gradient flow) is an important question, wherein the optimal choice of the learning rate appears to be an intriguing question, especially in the context of our numerical studies (c.f. Fig. 3). Finally, it would be of great interest to develop a general theory of SNNs for applications of neural network-based techniques to matrix learning problems.

## 7 REPRODUCIBILITY STATEMENT

In our paper, we dedicate substantial effort to improving the reproducibility and comprehensibility of both our theoretical results and numerical studies. We formally state and discuss the necessity and implications of our assumptions (please see the paragraphs following each assumption) before presenting our theoretical results. We also provide proof sketches of our main theoretical results. In these sketches, we present the key ideas and high-level directions and refer the reader to more detailed and complete proofs in the Appendices. For the numerical studies, we provide details of different settings in Section 5 and Appendix E. The python code used to conduct our experiments is included in the supplementary material as a zip file and is also publicly available at `https://github.com/porichoy-gupto/spectral-neural-nets`.

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

# Supplementary Material

## Implicit regularization via Spectral Neural Networks and non-linear matrix sensing

## A    PROOF OF THEOREM 1

Before presenting the Proof of Theorem 1, we define the few notations. Let $A \in \mathbb{R}^{d_1 \times d_2}, d_1 < d_2$ be a rectangular matrix and $a \in \mathbb{R}^{d_1}$ be a vector. We let $A_{ij}$ denote the $(i, j)$-entry of $A$, $A_{i*}$ denote the $i$-th row of $A$, and $A_{*j}$ denote the $j$-th column of $A$. We define the following functions on the matrix $A$.

$$\text{vec} : \mathbb{R}^{d_1 \times d_2} \to \mathbb{R}^{d_1 d_2} \qquad\qquad \text{vec}(A) = \begin{bmatrix} A_{*1}^\top & \cdots & A_{*d_2}^\top \end{bmatrix}^\top \qquad (14)$$

$$\text{diag} : \mathbb{R}^{d_1 \times d_2} \to \mathbb{R}^{d_1} \qquad\qquad \text{diag}(A) = [A_{11} \dots A_{d_1 d_1}]^\top \qquad (15)$$

$$\text{Diag} : \mathbb{R}^{d_1} \to \mathbb{R}^{d_1 \times d_2} 2 \qquad \text{Diag}(a) = \begin{bmatrix} a_1 & \cdots & 0 & 0 & \cdots & 0 \\ \vdots & \ddots & \vdots & \vdots & \ddots & \vdots \\ 0 & \cdots & a_{d_1} & 0 & \cdots & 0 \end{bmatrix}. \qquad (16)$$

We are now ready to present the Proof of Theorem 1. We first recall the definition of $X$ in Eq. 3

$$X = \sum_{k=1}^{K} \alpha_k \Gamma(U_k V_k^\top),$$

where $\Gamma(\cdot)$ is a matrix-valued function that applies a non-linear scalar-valued function $\gamma(\cdot)$ on the matrix's singular values. Under Assumption 1, we can write $U_k$ and $V_k$ as

$$U_k V_k^\top = (\Phi \bar{U}_k \Psi^T G)(G^\top \bar{V}_k^\top \Psi^\top) = \Phi \bar{U}_k \bar{V}_k^\top \Psi^\top.$$

Since both $\bar{U}_k$ and $\bar{V}_k$ are diagonal matrices, their product $\bar{U}_k \bar{V}_k^\top$ is also diagonal. Consequently, we can write

$$\Gamma(U_k V_k^\top) = \Phi \gamma(\bar{U}_k \bar{V}_k^\top) \Psi^\top,$$

where $\gamma(\cdot)$ is applied entry-wise on the matrix $\bar{U}_k \bar{V}_k^\top$. We can now write the matrix $X$ as

$$X = \sum_{k=1}^{K} \alpha_k \Phi \gamma(\bar{U}_k \bar{V}_k^\top) \Psi^\top = \Phi \left( \sum_{k=1}^{K} \alpha_k \gamma(\bar{U}_k \bar{V}_k^\top) \right) \Psi^\top. \qquad (17)$$

For notational convenience, we define the following notations to be used throughout this section:

$$\overline{X} = \sum_{k=1}^{K} \alpha_k \gamma(\bar{U}_k \bar{V}_k^\top) \in \mathbb{R}^{d_1 \times d_2} : \text{Diagonal matrix containing the singular values of } X \qquad (18)$$

$$\bar{\mathbf{x}} = \text{diag}(\overline{X}) \in \mathbb{R}^{d_1} : \text{Vector containing the singular values of } X \qquad (19)$$

$$\mathbf{x} = \text{vec}(X) \in \mathbb{R}^{d_1 d_2} : \text{The matrix } X \text{ expressed as a vector} \qquad (20)$$

$$\Theta = \Psi \boxtimes \Phi \in \mathbb{R}^{d_1 d_2 \times d_1} : \text{Khatri–Rao product between } \Psi \text{ and } \Phi \qquad (21)$$

$$G = \frac{\partial \ell(X)}{\partial X} = -\sum_{j=1}^{m} (y_j - \langle A_j, X \rangle) A_j \in \mathbb{R}^{d_1 \times d_2} : \text{Gradient of } \ell(X) \text{ with respect to } X \qquad (22)$$

$$\mathbf{g} = \text{vec}(G) \in \mathbb{R}^{d_1 d_2} : \text{The gradient } G \text{ expressed as a vector} \qquad (23)$$

$$Z_k = U_k V_k^\top \in \mathbb{R}^{d_1 \times d_2} : \text{Product matrix between } U_k \text{ and } V_k^\top. \qquad (24)$$

The Khatri–Rao product in Eq. 21 is defined as follows: the columns of the matrix $\Theta$ are Kronecker products of the corresponding columns of $\Psi$ and $\Phi$. In other words, the $i$-th column of $\Theta$ can be

expressed as the vectorization of the outer product between the $i$-th column of $\Phi$ and the $i$-th column of $\Psi$, i.e.,

$$\Theta_{*i} = \text{vec}(\phi_i \psi_i^\top). \tag{25}$$

We shall see in the next paragraph that by leveraging the Khatri–Rao product, we can write the differentials of many quantities of interest compactly, facilitating the derivation of the gradient flow dynamics in Theorem 4.

Therefore, we can use the Khatri–Rao product to expand $\text{vec}(X)$ as follows:

$$X = \Phi \overline{X} \Psi^\top = \sum_{i=1}^{d_1} \overline{X}_{ii} \phi_i \psi_i^\top \tag{26}$$

$$\mathbf{x} = \text{vec}(X) = \sum_{i=1}^{d_1} \overline{X}_{ii} \text{vec}(\phi_i \psi_i^\top) = \sum_{i=1}^{d_1} \overline{X}_{ii} \Theta_{*i} = \Theta \overline{\mathbf{x}}. \tag{27}$$

From here, we can write the differential of $X$ as

$$d\mathbf{x} = \Theta d\overline{\mathbf{x}}. \tag{28}$$

Since $\overline{\mathbf{x}}_i$ is the $i$-th singular value of the matrix $X$, we can express the differential of $\overline{\mathbf{x}}_i$ as follows:

$$d\overline{\mathbf{x}}_i = \langle \phi_i \psi_i^\top, dX \rangle = \langle \phi_i \psi_i^\top, \alpha_k \gamma'(z_i^{(k)}) dZ_k \rangle = \langle \alpha_k \gamma'(z_i^{(k)}) \phi_i \psi_i^\top, dZ_k \rangle, \tag{29}$$

where the first equality is due to Eq. 3, and the second equality is due to $\alpha_k$ and $\gamma'(z_i^{(k)})$ being scalars. Notice that we can write the vector $\overline{\mathbf{x}}$ as a sum over its entries as follow

$$\overline{\mathbf{x}} = \sum_{i=1}^{d_1} \overline{\mathbf{x}}_i \cdot \mathbf{e}_i,$$

where $\mathbf{e}_i$ denote the $i$-th canonical basis vectors of $\mathbb{R}^{d_1}$. We have

$$d\overline{\mathbf{x}} = \sum_{i=1}^{d_1} d\overline{\mathbf{x}}_i \cdot \mathbf{e}_i = \sum_{i=1}^{d_1} \langle \alpha_k \gamma'(z_i^{(k)}) \phi_i \psi_i^\top, dZ_k \rangle \cdot \mathbf{e}_i = \alpha_k \gamma'(z_i^{(k)}) \Big\langle \sum_{i=1}^{d_1} \mathbf{e}_i \star \phi_i \psi_i^\top, dZ_k \Big\rangle, \tag{30}$$

where $\star$ denotes the tensor product, i.e., $\mathbf{e}_i \star \phi_i \psi_i^\top \in \mathbb{R}^{d_1 \times d_1 \times d_2}$ is a third-order tensor. Since $dZ_k$ has dimension $d_1 \times d_2$, the above Frobenius product returns a vector of dimension $d_1$, which matches that of $d\overline{\mathbf{x}}$. Substituting Eq. 30 this into the differential of $\ell(X)$ gives

$$d\ell(X) = \Big\langle \frac{\partial \ell}{\partial X}, dX \Big\rangle = \langle G, dX \rangle = \mathbf{g}^\top d\mathbf{x} = \mathbf{g}^\top \Theta d\overline{\mathbf{x}}$$

$$= \alpha_k \gamma'(z_i^{(k)}) \Big\langle \sum_{i=1}^{d_1} (\mathbf{g}^\top \Theta \mathbf{e}_i)(\phi_i \psi_i^\top), dZ_k \Big\rangle. \tag{31}$$

Let us define the scalar $\lambda_i^{(k)}$ as

$$\lambda_i^{(k)} = -\alpha_k \gamma'(z_i^{(k)}) \, \mathbf{g}^\top \Theta \mathbf{e}_i$$

$$\overset{(a)}{=} -\alpha_k \gamma'(z_i^{(k)}) \, \mathbf{g}^\top \Theta_{*i}$$

$$\overset{(b)}{=} -\alpha_k \gamma'(z_i^{(k)}) \, \mathbf{g}^\top \text{vec}(\phi_i \psi_i^\top)$$

$$= -\alpha_k \gamma'(z_i^{(k)}) \langle G, \phi_i \psi_i^\top \rangle$$

$$\overset{(c)}{=} \alpha_k \gamma'(z_i^{(k)}) \Big\langle \sum_{j=1}^{m} (y_j - \langle A_j, X \rangle) A_j, \phi_i \psi_i^\top \Big\rangle$$

$$= \alpha_k \gamma'(z_i^{(k)}) \sum_{j=1}^{m} (y_j - \langle A_j, X \rangle) \Big\langle A_j, \phi_i \psi_i^\top \Big\rangle$$

$$\stackrel{(d)}{=} \alpha_k \gamma'(z_i^{(k)}) \sum_{j=1}^{m} \left( \langle \sigma^{(j)}, \sigma^\star \rangle - \sum_{l=1}^{K} \alpha_l \langle \sigma^{(j)}, \gamma(z^{(l)}) \rangle \right) \cdot \sigma_i^{(j)}$$

$$\stackrel{(e)}{=} \alpha_k \gamma'(z_i^{(k)}) \sum_{j=1}^{m} \left( B^\top \sigma^\star - B^\top H \alpha \right) \cdot \sigma_i^{(j)}$$

$$= \alpha_k \gamma'(z_i^{(k)}) \cdot \mathrm{row}_i(B) B^\top (\sigma^\star - H\alpha),$$

where $(a)$ is due to $\Theta \mathbf{e}_i$ equals to the $i$-column of $\Theta$, $(b)$ is due to Eq. 25, $(c)$ is due to the definition of the matrix $G$ in Eq. 22, $(d)$ is due to Eq. 10, and $(e)$ is due to the definitions of $B$ and $H$.

Let $\lambda^{(k)} \in \mathbb{R}^{d_1}$ denote the vector containing the $\lambda_i^{(k)}$, we can write

$$\lambda^{(k)} = \alpha_k \cdot \gamma'(z^{(k)}) \circ BB^\top (\sigma^\star - H\alpha) = \alpha_k \cdot \gamma'(z^{(k)}) \circ \mathcal{C}(\sigma^\star - H\alpha). \tag{32}$$

The differential of $d\ell(X)$ becomes

$$d\ell(X) = -\left\langle \sum_{i=1}^{d_1} \lambda_i^{(k)} \phi_i \psi_i^\top, dZ_k \right\rangle,$$

$$\frac{\partial \ell(X)}{\partial Z_k} = -\sum_{i=1}^{d_1} \lambda_i^{(k)} \phi_i \psi_i^\top = -\Phi L^{(k)} \Psi^\top, \tag{33}$$

where $L^{(k)}$ is a diagonal matrix whose diagonal entries are $\lambda_i^{(k)}$, i.e., $L^{(k)} = \mathrm{Diag}(\lambda^{(k)})$. Since $Z_k = U_k V_k^\top$, we have

$$\frac{\partial \ell(X)}{\partial U_k} = -\sum_{i=1}^{d_1} \lambda_i^{(k)} \phi_i \psi_i^\top = -\Phi L^{(k)} \Psi^\top V_k, \tag{34}$$

$$\frac{\partial \ell(X)}{\partial V_k} = -\sum_{i=1}^{d_1} \lambda_i^{(k)} \phi_i \psi_i^\top = -(\Phi L^{(k)} \Psi^\top)^\top U_k. \tag{35}$$

This concludes the proof for the gradient flow dynamics on $U_k$ and $V_k$. In the remaining, we shall derive the gradient of $\ell(X)$ with respect to the scalar $\alpha_k$.

$$\frac{\partial \ell(X)}{\partial \alpha_k} = -\sum_{j=1}^{m} (y_j - \langle A_j, X \rangle) \langle \Gamma(U_k V_k^\top), A_j \rangle$$

$$= -\sum_{j=1}^{m} \left( \langle \sigma^{(i)}, \sigma^\star - \sum_{l=1}^{K} \alpha_l \gamma(z^{(l)}) \rangle \cdot \langle \sigma^{(i)}, \gamma(z^{(k)}) \rangle \right)$$

$$= -\gamma(z^{(k)})^\top \mathcal{C} \left( \sigma^\star - \sum_{l=1}^{K} \alpha_l \gamma(z^{(l)}) \right)$$

$$= -\gamma(z^{(k)})^\top \mathcal{C} (\sigma^\star - H\alpha),$$

where the second equality is due to Eq. 10. Consequently, the gradient of $\ell(X)$ with respect to the vector $\alpha$ is

$$\frac{\partial \ell(X)}{\partial \alpha} = -H^\top \mathcal{C} (\sigma^\star - H\alpha), \tag{36}$$

which concludes the proof of Theorem 1.

## B  PROOF OF THEOREM 2

Let us direct our attention to the evolution of the diagonal elements. Restricting 12 to the diagonal elements gives us a system of differential equations for each $i \in [m]$:

$$[\partial_t \bar{U}_k]_{ii} = \lambda_i^{(k)} [\bar{V}_k]_{ii}, \quad [\partial_t \bar{V}_k]_{ii} = \lambda_i^{(k)} [\bar{U}_k]_{ii}. \tag{37}$$

We can re-write the above into a single matrix differential equation as

$$\partial_t \begin{bmatrix} [\bar{U}_k]_{ii} \\ [\bar{V}_k]_{ii} \end{bmatrix} = \lambda_i \begin{bmatrix} 0 & 1 \\ 1 & 0 \end{bmatrix} \begin{bmatrix} [\bar{U}_k]_{ii} \\ [\bar{V}_k]_{ii} \end{bmatrix}. \tag{38}$$

For the remaining of this section, we define the following notations for ease of presentation:

$$\mathbf{x}_i^{(k)} = \begin{bmatrix} [\bar{U}_k]_{ii} \\ [\bar{V}_k]_{ii} \end{bmatrix} \tag{39}$$

$$A = \begin{bmatrix} 0 & 1 \\ 1 & 0 \end{bmatrix} \tag{40}$$

$$w_i^{(k)} = \frac{1}{2}\mathbf{x}_i^{(k)\top}\mathbf{x}_i^{(k)} = \frac{1}{2}(\bar{U}_{ii}^2 + \bar{V}_{ii}^2) \tag{41}$$

$$\partial_t w_i^{(k)} = \mathbf{x}_i^{(k)\top}\partial_t\mathbf{x}_i^{(k)} \tag{42}$$

$$z_i^{(k)} = \frac{1}{2}\mathbf{x}_i^{(k)\top}A\mathbf{x}_i^{(k)} = [\bar{U}_k]_{ii}[\bar{V}_k]_{ii} \tag{43}$$

$$\partial_t z_i^{(k)} = \mathbf{x}_i^{(k)\top}A\partial_t\mathbf{x}_i^{(k)}. \tag{44}$$

The above matrix differential equation becomes

$$\partial_t\mathbf{x}_i^{(k)} = \lambda_i^{(k)}A\mathbf{x}_i^{(k)}$$
$$\mathbf{x}_i^{(k)\top}\partial_t\mathbf{x}_i^{(k)} = \mathbf{x}_i^{(k)\top}\lambda_i^{(k)}A\mathbf{x}_i^{(k)}$$
$$\partial_t w_i^{(k)} = 2\lambda_i^{(k)}z_i^{(k)} \tag{45}$$

On another note, we also have

$$\partial_t\mathbf{x}_i^{(k)} = \lambda_i^{(k)}A\mathbf{x}_i^{(k)}$$
$$\mathbf{x}_i^{(k)\top}A\partial_t\mathbf{x}_i^{(k)} = \lambda_i^{(k)}\mathbf{x}_i^{(k)\top}AA\mathbf{x}_i^{(k)}$$
$$\partial_t z_i^{(k)} = 2\lambda_i^{(k)}w_i^{(k)}. \tag{46}$$

We are now ready to prove the main result. In the remaining proof, we will derive the differential equation for $H\alpha$. By the product rule of calculus, we have

$$\partial_t(H\alpha) = (\partial_t H)\alpha + H(\partial_t\alpha). \tag{47}$$

We shall derive $\partial_t H\alpha$ and $H\partial_t\alpha$ separately. First, let us consider the evolution of $H$ over time.

$$\partial_t H = \begin{bmatrix} \ldots \,|\, \gamma'(z^{(k)})\partial_t z^{(k)} \,|\, \ldots \end{bmatrix} \tag{48}$$

$$= \begin{bmatrix} \ldots \,|\, \gamma'(z^{(k)}) \circ 2\lambda^{(k)} \circ w^{(k)} \,|\, \ldots \end{bmatrix} \tag{49}$$

$$= \begin{bmatrix} \ldots \,|\, \gamma'(z^{(k)})^2 \circ 2\mathcal{C}(\sigma^\star - H\alpha) \circ w^{(k)} \,|\, \ldots \end{bmatrix} \mathrm{Diag}(\alpha), \tag{50}$$

where the first equality follows from the definition of $H$ and the chain rule of calculus, the second equality is due to Eq. 46, and the last equality follows from Theorem 1. Multiplying the vector $\alpha$ from the right on both sides gives:

$$(\partial_t H)\alpha = 2\sum_{k=1}^{K}\alpha_k^2 \cdot \left(\gamma'(z^{(k)})^2 \circ \mathcal{C}(\sigma^\star - H\alpha) \circ w^{(k)}\right). \tag{51}$$

Recall that from Theorem 1, we have

$$\partial_t\alpha = H^\top\mathcal{C}(\sigma^\star - H\alpha). \tag{52}$$

Multiplying the matrix $H$ from the left on both sides gives

$$H(\partial_t\alpha) = HH^\top\mathcal{C}(\sigma^\star - H\alpha). \tag{53}$$

Combining Eq. 51 and Eq. 53 gives

$$\partial_t(H\alpha) = 4\sum_{k=1}^{K}\alpha_k^2 \cdot \left(\gamma'(z^{(k)})^2 \circ \mathcal{C}(\sigma^\star - H\alpha) \circ w^{(k)}\right) + HH^\top\mathcal{C}(\sigma^\star - H\alpha). \tag{54}$$

Notice that by definition, the matrix $B$ contains the singular values of $A_i$'s, and therefore its entries are non-negative. Consequently, since $\mathcal{C} = BB^\top$, $\mathcal{C}$'s entries are also non-negative. Finally, by definition in Eq. 41, $w^{(k)}$ has non-negative entries. Therefore, all quantities in Eq. 54 are non-negative entry-wise, except for the vectors $(\sigma^\star - H\alpha)$. Consequently, both quantities $(\partial_t H)\alpha$ and $H(\partial_t \alpha)$ have the same sign as $(\sigma^\star - H\alpha)$. By our initialization, this sign is non-negative.

Furthermore, this non-negativity implies that

$$\partial_t(H\alpha) \geq HH^\top \mathcal{C}(\sigma^\star - H\alpha). \tag{55}$$

$$\partial_t(H\alpha) \geq 4\sum_{k=1}^{K} \alpha_k^2 \cdot \big(\gamma'(z^{(k)})^2 \circ \mathcal{C}(\sigma^\star - H\alpha) \circ w^{(k)}\big). \tag{56}$$

We will have the occasion to use both inequalities depending on the situation. Finally, we can also write down a similar differential equation for each $(H_{ij}$ from Eq. 50 as

$$\partial_t H_{ij} = 2 \cdot \alpha_j \gamma'(z_i^{(j)})^2 [\mathcal{C}(\sigma^\star - H\alpha)]_{ij} w_i^{(k)}. \tag{57}$$

By Assumption 2, part (c), at initialization, we have $H(0)\alpha(0) < \sigma^\star$ entry-wise. This implies that each entry in $\partial_t(H\alpha)$ is positive at initialization, and therefore $H\alpha$ is increasing in a neighborhood of 0. As $H\alpha$ approaches $\sigma^\star$, the gradient $\partial_t(H\alpha)$ decreases and reaches 0 exactly when $H\alpha = \sigma^\star$, which then causes $H\alpha$ to stay constant from then on. Thus, we have shown that

$$\lim_{t\to\infty} H(t)\alpha(t) = \sigma^\star \quad \text{and} \quad \partial_t(H\alpha) \geq 0. \tag{58}$$

Combining Eq. 46 and Eq. 45, we have

$$\partial_t w_i^{(k)} \cdot w_i^{(k)} - \partial_t z_i^{(k)} \cdot z_i^{(k)} = 0. \tag{59}$$

Integrating both sides with respect to time, we have that for any $t > 0$

$$(w_i^{(k)}(t))^2 - (z_i^{(k)}(t))^2 = Q. \tag{60}$$

for some constant $Q$ which does not depend on time. Since $w_i^{(k)}$ is non-negative by definition, the above implies that $w_i^{(k)}(t) \geq \sqrt{|Q|}$ for all $t > 0$; note that $Q > 0$ can be ensured via initialization, as discussed below. To this end, notice that

$$(w_i^{(k)}(0))^2 - (z_i^{(k)}(0))^2 = \frac{1}{4}(\bar{U}_{k,ii}(0)^2 + \bar{V}_{k,ii}(0)^2)^2 - \bar{U}_{k,ii}(0)^2 \bar{V}_{k,ii}(0)^2$$

$$= \frac{1}{4}(\bar{U}_{k,ii}(0)^2 - \bar{V}_{k,ii}(0)^2)^2.$$

Thus this can always be arranged simply by initializing $\bar{U}_k(0), \bar{V}_k(0)$ suitably.

This, in particular, implies that $w_i^{(k)}$ is bounded away from 0 in time.

In the remainder of this section, we will show that the convergence rate is exponential. In the below, we shall establish a lower bound on $\partial_t(H\alpha)$.

**Case 1:** The $\alpha_k$ are upper bounded by a finite constant $\alpha_{\max} > 0$ for all $k \in [K]$ and $t \in \mathbb{R}_+$.

Let $h_1, \ldots, h_{d_1}$ denote the rows of $H$, and $b_1, \ldots, b_{d_1}$ denote the rows of $B$. We have

$$H\partial_t\alpha = HH^\top \mathcal{C}(\sigma^\star - H\alpha) = (HH^\top)(BB^\top)(\sigma^\star - H\alpha) \tag{61}$$

$$= \begin{bmatrix} h_1^\top h_1 & h_1^\top h_2 & \cdots & h_1^\top h_{d_1} \\ \vdots & \vdots & \ddots & \vdots \\ h_1^\top h_{d_1} & h_2^\top h_{d_1} & \cdots & h_{d_1}^\top h_{d_1} \end{bmatrix} \begin{bmatrix} b_1^\top b_1 & b_1^\top b_2 & \cdots & b_1^\top b_{d_1} \\ \vdots & \vdots & \ddots & \vdots \\ b_1^\top b_{d_1} & b_2^\top b_{d_1} & \cdots & b_{d_1}^\top b_{d_1} \end{bmatrix} (\sigma^\star - H\alpha) \tag{62}$$

$$\geq \begin{bmatrix} h_1^\top h_1 & & & \\ & h_2^\top h_2 & & \\ & & \ddots & \\ & & & h_{d_1}^\top h_{d_1} \end{bmatrix} \begin{bmatrix} b_1^\top b_1 & & & \\ & b_2^\top b_2 & & \\ & & \ddots & \\ & & & b_{d_1}^\top b_{d_1} \end{bmatrix} (\sigma^\star - H\alpha) \tag{63}$$

$$= \begin{bmatrix} h_1^\top h_1 \cdot b_1^\top b_1 & & & \\ & h_2^\top h_2 \cdot b_2^\top b_2 & & \\ & & \ddots & \\ & & & h_{d_1}^\top h_{d_1} \cdot b_{d_1}^\top b_{d_1} \end{bmatrix} (\sigma^\star - H\alpha) \tag{64}$$

$$= \begin{bmatrix} h_1^\top h_1 \cdot b_1^\top b_1 & \cdots & h_{d_1}^\top h_{d_1} \cdot b_{d_1}^\top b_{d_1} \end{bmatrix}^\top \circ (\sigma^\star - H\alpha), \tag{65}$$

where the first inequality is due to the non-negativity of the entries of $H$ and $B$. Let us focus our attention on the evolution of the $i$-th entry of $H\alpha$.

$$\partial_t (H\alpha)_i = [(\partial_t H)\alpha]_i + [H(\partial_t \alpha)]_i \geq [H(\partial_t \alpha)]_i \geq (h_i^\top h_i \cdot b_i^\top b_i)(\sigma_i^\star - (H\alpha)_i), \tag{66}$$

where the first inequality is due to $[H(t)\alpha(t)]_i \leq \sigma_i^\star$, which causes $(\partial_t H)\alpha_i$ to be non-negative.

Notice that $b_i$ are constants with respect to time, and are non-zero because of the condition that the all-ones vector lies in the range of $B$. Therefore, to show a lower bound on $\partial_t (H\alpha)_i$, it remains to show that $h_i^\top h_i$ (or $\|h_i\|$) is bounded away from 0.

In the previous part, we have shown that $H(t)\alpha(t)$ approaches $\sigma^\star$ as $t \to \infty$. Therefore, for any $\epsilon > 0$, there exists a time $t_0$ after which $|\sigma_i^\star - (H\alpha)_i| \leq \epsilon$. Notice that $(H\alpha)_i = h_i^\top \alpha$. By Cauchy-Schwarz inequality, we have for $t > t_0$:

$$\|h_i(t)\| \cdot \|\alpha(t)\| \geq |h_i(t)^\top \alpha(t)| \geq |\sigma_i^\star| - |\sigma_i^\star - h_i(t)^\top \alpha(t)| \geq |\sigma_i^\star| - \epsilon, \tag{67}$$

where the second inequality is due to the Triangle inequality. Choose $\epsilon = |\sigma_i^\star|/2$, we have

$$\|h_i(t)\| \geq \frac{|\sigma_i^\star|}{2\|\alpha(t)\|} \geq \frac{|\sigma_i^\star|}{2\sqrt{K}\alpha_{\max}}. \tag{68}$$

Let us define the constant $\eta_i = |\sigma_i^\star|/(2\sqrt{K}\alpha_{\max}) \cdot b_i^\top b_i$, and $\beta_i = \sigma_i^\star - (H\alpha)_i$. Notice that $\eta_i$ is a constant with respect to $t$. We have the following differential inequality:

$$\partial_t \beta_i = -\partial_t (H\alpha)_i \leq -\eta_i \beta_i. \tag{69}$$

Integrating the above differential inequality we get that

$$\beta_i(t) = \sigma_i^\star - (H(t)\alpha(t))_i \leq Ce^{-\eta_i t}, \tag{70}$$

for some constant $C > 0$ for all large enough $t$. This shows that $H\alpha$ converges to $\sigma^\star$ at an exponential rate.

**Definition.** In the complement of Case 1, we define the subset $S \subseteq [K]$ such that $j \in S$ implies that $\lim_{t\to\infty} \alpha_j(t) = +\infty$.

In order to deal with the complement of Case 1 above, we now proceed to handle the convergence of $H\alpha$ to $\sigma^*$ coordinate-wise. To this end, we consider two types of coordinates $i \in [d_1]$, depending on the limiting behavior of the $H_{ij}$ in tandem with that of $\alpha_j$ (as $j$ ranges over $S$ for this $i$).

**Case 2 :** The index $i \in [d_1]$ is such that $H_{ij}\alpha_j \to 0$ as $t \to \infty$ for all $j \in S$. In this case, we consider the left and right sides of Eq 54 for the $i$-th co-ordinate, and notice that in fact we have $\sigma_i^* = \lim_{t\to\infty} \sum_{j\notin S} H_{ij}\alpha_j$. Since $\alpha_j$ for each $j \notin S$ converges to a finite real number, this implies that for this particular index $i \in [d_1]$ we can employ the argument of Case 1, with a lower dimensional vector $(\alpha_j)_{j\notin S}$ instead of $(\alpha_j)_{j\in[K]}$.

**Case 3 :** The index $i \in [d_1]$ is such that $\exists j \in S$ with $H_{ij}\alpha_j$ does not converge to zero as $t \to \infty$ and $H_{ij}$ remains bounded away from 0 for this $j$. In this case, we notice that the $i$-th row of $H$, denoted $h_i$, satisfies the condition that $\|h_i\|_2$ is bounded away from 0 as $t \to \infty$, thanks to the above co-ordinate $j$. As such, we are able to apply the exponential decay argument of Case 1 by combining Eq 66 and Eq 69.

**Case 4 :** The index $i \in [d_1]$ is such that $\exists j \in S$ with $H_{ij}\alpha_j$ does not converge to zero as $t \to \infty$ and $H_{ij} \to 0$ for this $j$. For the proof of Case 4 in Theorem 2, we further assume that if $x_0$ is a zero of the activation function $\gamma$, then $\gamma'(x) \geq c\gamma(x)$ for $x$ sufficiently close to $x_0$, and $c >$ being a constant. This covers the case of $x_0 = -\infty$, as in the case of sigmoid functions, where the condition would be taken to be satisfied for $x$ sufficiently large and negative. This condition is satisfied by nearly all of

the activation functions of our interest, including sigmoid functions, the tanh function, and truncated ReLu with quadratic smoothing, among others. We believe that this condition is a technical artifact of our proof, and endeavor to remove it in an upcoming version of the manuscript.

In this setting, we invoke Eq 54 in its $i$-th component and lower bound its right-hand side by the $k = j$ summand therein; in other words, we write

$$\partial_t (H\alpha)_i \geq 4\alpha_j^2 \gamma'(z_i^{(j)})^2 [\mathcal{C}(\sigma^\star - H\alpha)]_i w_i^{(k)}. \tag{71}$$

Since $H_{ij} = \gamma(z_i^{(j)})$, therefore for large enough $t$, we have $z_i^{(j)}$ is close to $z_0$, the zero of $\gamma$ (in the event $z_0 = -\infty$, this means that $z_i^{(j)}$ is large and negative). By the properties of the activation function $\gamma$, this implies that the inequality $\gamma'(z_i^{(j)}) \geq c\gamma(z_i^{(j)})$ holds true with an appropriate constant $c > 0$ for large enough $t$. Notice that $\alpha_j \uparrow +\infty$ and from Eq (57) we have $\partial_t H_{ij} \geq 0$ for large enough time $t$, which implies that $H_{ij}$ is non-decreasing for large time. As a result, $H_{ij}\alpha_j$ is non-decreasing for large time. But recall that $H_{ij}\alpha_j$ does not converge to zero as $t \to \infty$ by the defining condition of this case, which when combined with the non-decreasing property established above, implies that $H_{ij}\alpha_j$ is bounded away from 0 for large time. Finally, recall that $w_i^k \geq \sqrt{|Q|}$. Combining these observations, we may deduce from Eq 72 that, for an appropriate constants $c_1, c_2 > 0$ we have

$$\partial_t (H\alpha)_i \geq c_1 \alpha_j^2 \gamma(z_i^{(j)})^2 [\mathcal{C}(\sigma^\star - H\alpha)]_i w_i^{(k)} \geq c_2 \cdot [\mathcal{C}(\sigma^\star - H\alpha)]_i. \tag{72}$$

We now proceed as in Eq 69 and obtain exponentially fast convergence, as desired.

Establishing a matching exponential lower bound on $\beta_i(t)$ is not very important from a practical point of view. So, we only show such a bound under the assumption that the $\alpha_k(t)$ remains bounded by some constant $\alpha_{\max} > 0$, for all $k$. For simplicity, we also assume that the non-linearity $\gamma$ is such that $\gamma'(x) = O\left(\frac{1}{\sqrt{|x|}}\right)$ as $x \to \pm\infty$. (This is a mild assumption, satisfied, e.g., by the logistic or the tanh non-linearities.) Since $w_i^{(k)}$ can grow at most linearly in $z_i^{(k)}$ (see Eq. (60)), this assumption ensures that $\gamma'(z_i^{(k)})^2 w_i^{(k)}$ remains bounded uniformly for all $i$ and $k$. Further, the entries of $H$ are uniformly bounded, and $\mathcal{C}$ is a fixed matrix. Therefore, for some constant $\zeta > 0$, we have, for all $i \in [d_1]$, that

$$\partial_t (H\alpha)_i = 4\sum_{k=1}^{K} \alpha_k^2 (\gamma'(z_i^{(k)}))^2 w_i^{(k)} (\mathcal{C}(\sigma^\star - H\alpha)_i + \sum_j (HH^\top)_{ij} (C(\sigma^\star - H\alpha))_j$$
$$\leq \zeta \sum_\ell (\sigma^\star - H\alpha)_\ell.$$

A fortiori,

$$\partial_t \sum_i \beta_i = \sum_i \partial_t \beta_i = -\sum_i \partial_t (H\alpha)_i$$
$$\geq -\zeta \sum_i \sum_\ell \beta_\ell$$
$$= -d_1 \zeta \sum_i \beta_i.$$

Integrating this differential inequality, we conclude that for some constant $C_1 > 0$,

$$\sum_i \beta_i(t) \geq C_1 e^{-d_1 \zeta t},$$

for all $t > 0$. The Cauchy-Schwartz bound $\sum_i \beta_i(t) \leq \sqrt{d_1} \|\beta(t)\|_2$ then implies that for all $t > 0$,

$$\|\sigma^\star - H(t)\alpha(t)\|_2 = \|\beta(t)\|_2 \geq \frac{C_1}{\sqrt{d_1}} e^{-d_1 \zeta t} = C e^{-\eta t},$$

where $C = \frac{C_1}{\sqrt{d_1}}$ and $\eta = d_1 \zeta$. This completes the proof of the lower bound.

## C  PROOF OF THEOREM 3

By our definition of $X_\infty$, we have

$$
\begin{aligned}
X_\infty &= \lim_{t\to\infty} X(t) \\
&= \lim_{t\to\infty} \sum_{k=1}^{K} \alpha_k(t) \Gamma\big(U_k(t) V_k(t)^\top\big) \\
&= \lim_{t\to\infty} \sum_{k=1}^{K} \alpha_k(t) \Gamma\big(\Phi \bar{U}_k(t) G \cdot G^\top \bar{V}_k(t)^\top \Psi^\top\big) \\
&= \lim_{t\to\infty} \sum_{k=1}^{K} \alpha_k(t) \Gamma\big(\Phi \bar{U}_k(t) \bar{V}_k(t)^\top \Psi^\top\big) \\
&= \lim_{t\to\infty} \sum_{k=1}^{K} \alpha_k(t) \Phi \gamma(\bar{U}_k(t) \bar{V}_k(t)^\top \Psi^\top \\
&= \Phi \bigg[ \lim_{t\to\infty} \sum_{k=1}^{K} \alpha_k(t) \gamma\big(\bar{U}_k(t) \bar{V}_k(t)^\top\big) \bigg] \Psi^\top \\
&= \Phi \bigg[ \lim_{t\to\infty} \mathrm{Diag}\Big( \sum_{k=1}^{K} \alpha_k(t) \gamma\big(z^{(k)}(t)\big) \Big) \bigg] \Psi^\top \\
&= \Phi \bigg[ \lim_{t\to\infty} \mathrm{Diag}\Big( H(t)\boldsymbol{\alpha}(t) \Big) \bigg] \Psi^\top \\
&= \Phi \bigg[ \mathrm{Diag}\Big( \lim_{t\to\infty} H(t)\boldsymbol{\alpha}(t) \Big) \bigg] \Psi^\top \\
&= \Phi \Big[ \mathrm{Diag}(\sigma^\star) \Big] \Psi^\top,
\end{aligned}
$$

where the fourth equality is due to the orthogonality of $G$, and the last equality is due to Theorem 1. Now, let us look at the prediction given by $X_\infty$, for any $i = 1, \ldots, m$, we have

$$
\langle A_i, X_\infty \rangle = \langle \Phi \bar{A}_i \Psi^\top, \Phi \mathrm{Diag}(\sigma^\star) \Psi^\top \rangle = \langle \sigma^{(i)}, \sigma^\star \rangle = y_i, \tag{73}
$$

where the last equality is due to Equation 10. This implies that $\ell(X_\infty) = 0$, proving (a).

To prove (b), we will show that $X_\infty$ satisfies the Karush-Kuhn-Tucker (KKT) conditions of the following optimization problem

$$
\min_{X \in \mathbb{R}^{d_1 \times d_2}} \|X\|_* \quad \text{subject to} \quad y_i = \langle A_i, X \rangle, \forall i \in [m]. \tag{74}
$$

The KKT optimality conditions for the optimization in 74 are

$$
\exists \nu \in \mathbb{R}^m \quad \text{s.t.} \quad \forall i \in [m], \quad \langle A_i, X \rangle = y_i \quad \text{and} \quad \nabla_X \|X\|_* + \sum_{i=1}^{m} \nu_i A_i = 0. \tag{75}
$$

The solution matrix $X_\infty$ satisfies the first condition from the first claim of Theorem 3, it remains to prove that $X_\infty$ also satisfies the second condition. The gradient of the nuclear norm of $X_\infty$ is given by

$$
\nabla \|X_\infty\| = \Phi \Psi^\top. \tag{76}
$$

Therefore, the second condition becomes

$$
\Phi \Psi^\top + \sum_{i=1}^{m} \nu_i A_i = 0
$$

$$
\Leftrightarrow \quad \Phi \Psi^\top + \sum_{i=1}^{m} \nu_i \Phi \bar{A}_i \Psi^\top = 0
$$

$$\Leftrightarrow \quad \Phi\Big(I - \sum_{i=1}^{m} \nu_i \bar{A}_i\Big)\Psi^\top = 0$$

$$\Leftrightarrow \quad \sum_{i=1}^{m} \nu_i \sigma_j^{(i)} = 1, \forall j \in [d_1]$$

$$\Leftrightarrow \quad B\nu = \mathbb{1}.$$

By Assumption 1, the vector $\mathbb{1}$ lies in the column space of $B$, which implies the existence of a vector $\nu$ that satisfies the condition above. This proves (b) and concludes the proof of Theorem 3.

## D   SINGULAR VALUE DECOMPOSITION

In this appendix, we explain in detail the singular value decomposition of a rectangular matrix. By doing so, we also explain some of the non-standard notations used in the paper (e.g., a rectangular diagonal matrix).

In our paper, we consider a rectangular measurement matrix $A$ of dimension $d_1 \times d_2$ and rank $r$. Without loss of generality, we assume that $r \leq d_1 \leq d_2$. Then the singular value decomposition of the matrix $A$ is given by

$$A = \Phi\bar{A}\Psi^\top, \tag{77}$$

where $\Phi \in \mathbb{R}^{d_1 \times d_1}$, $\Psi \in \mathbb{R}^{d_2 \times d_1}$ are two orthogonal matrices whose columns represent the left and right singular vectors of $A$, and $\bar{A} \in \mathbb{R}^{d_1 \times d_1}$ is a diagonal matrix whose diagonal entries represent the singular values of $A$.

This is similar to the notion of *compact SVD* commonly used in the literature, but we truncate the matrix $\Psi$ to $d_1$ columns, instead of $r$ columns like in compact SVD.

This choice of compact SVD is not taken lightly, it is crucial for the proof of Theorem 1 via the use of the Khatri–Rao product. More specifically, the Khatri–Rao product in Eq. 21 requires that $\Phi$ and $\Psi$ have the same number of columns. This is generally not true in the standard singular value decomposition.

Next, we precisely define the notion of a rectangular diagonal matrix. Suppose $A \in \mathbb{R}^{d_1 \times d_2}$ with $d_1 < d_2$ is a diagonal matrix, then $A$ takes the following form:

$$A = \begin{bmatrix} a_1 & \cdots & 0 & 0 & \cdots & 0 \\ \vdots & \ddots & \vdots & \vdots & \ddots & \vdots \\ 0 & \cdots & a_{d_1} & 0 & \cdots & 0 \end{bmatrix}. \tag{78}$$

Here, we start with a standard square diagonal matrix of size $d_1 \times d_1$, and add $d_2 - d_1$ columns of all 0's to the right of that matrix. Similarly, we define a rectangular identity matrix of size $d_1 \times d_2$ by setting $a_1 = \cdots = a_{d_1} = 1$.

## E   ADDITIONAL NUMERICAL STUDIES

In this section, we complement the numerical studies in the main paper with some simulation results on the SNN architecture of Fig. 1. We empirically demonstrate that gradient descent dynamics in SNN minimizes the nuclear norm of the solution matrix, and leads to favorable generalization performance. We also investigate some settings in which the assumptions made in our theoretical study are relaxed.

**Do the results in Theorem 2 and Theorem 3 extend to multi-layer SNN?**

We generate the true matrix by sampling each entry of $X^\star$ independently from a standard Gaussian distribution, suitably normalized. For every measurement matrix $A_i$, $i = 1, \ldots, m$, we sample each entry of the diagonal matrix $\bar{A}_i$ from the uniform distribution on $(0, 1)$, sort them in decreasing order, and set $A_i = \Phi\bar{A}_i\Psi^\top$, where $\Phi$ and $\Psi$ are randomly generated orthogonal matrices. We then record the measurements $y_i = \langle A_i, X^\star \rangle$, $i = 1, \ldots, m$. A total of 120 measurement matrices are generated, half of them are used for training and the other half are used for testing ($m = 60$ in this case).

We use a two-layer SNN of size $[5, 3, 1]$, i.e., the input layer has 5 blocks, the hidden layer has 3 blocks, and the output layer has 1 block. These blocks are initialized with spectral initialization (see Assumption 2).

We take $d_1 = d_2 = 10$ (thus $X^\star$ has 100 entries), $m = 60$ measurement matrices, and $K = 3$. As for the non-linearity, we use the shifted and scaled sigmoid

$$\gamma(x) = \left[ \frac{1}{1 + e^{-x}} - \frac{1}{2} \right] * 0.8. \tag{79}$$

In the figures below, we demonstrate the evolution of the nuclear norm, the train error, and the test error of our SNN over time as we run gradient descent. Even though we only analyze gradient flow dynamics for a single block in our theoretical investigation, the empirical results suggest that the same phenomenon extends to more general multi-layer SNN.

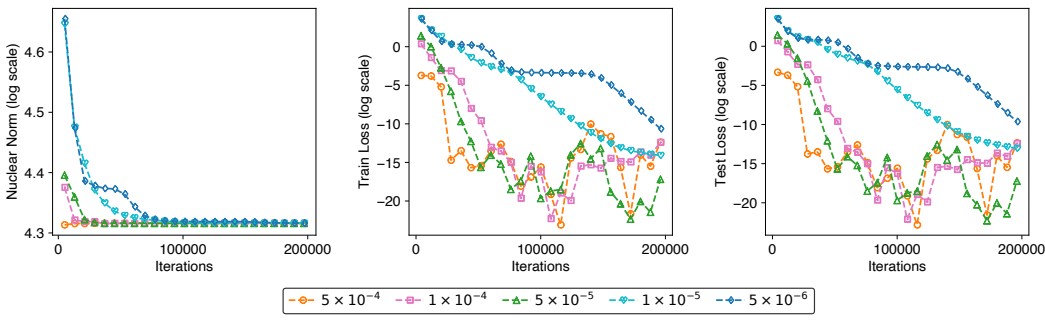

Figure 4: The evolution of SNN's nuclear norm, train error, and test error over time with learning rates (LR) of different magnitudes. In the left plot, the nuclear norm decays over time and converges to the same minimum value regardless of the learning rates.

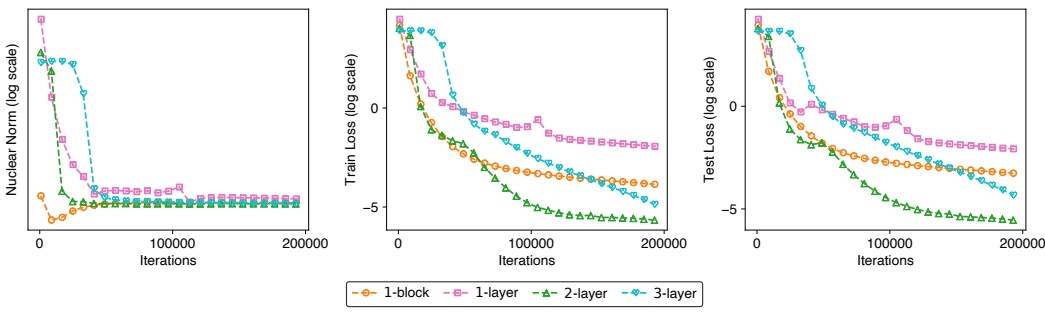

Figure 5: The evolution of SNN's nuclear norm, train error, and test error over time with various SNN architectures. We use a learning rate of $10^{-5}$. The left plot shows that gradient descent minimizes the nuclear norm of the solution matrix for all architectures, although they converge to slightly different final solutions. In the right plot, we observe that there is a substantial difference in the test error between different architectures. In this particular simulation, a 2-layer SNN achieves the lowest test error. This hints at the advantage of going beyond a single block to multi-layer SNN architecture as we proposed in Section 3.

**Do the results in Theorem 2 and Theorem 3 still hold when we relax the assumptions?**

We investigate a scenario in which Assumption 1 (commuting measurement matrices) and Assumption 2 (spectral initialization) are relaxed.

We generate the true matrix by sampling each entry of $X^\star$ independently from a standard Gaussian distribution, suitably normalized. For every measurement matrix $A_i$, $i = 1, \ldots, m$, we sample each entry of $A_i$ from a standard Gaussian distribution. We then record the measurements $y_i = \langle A_i, X^\star \rangle$, $i = 1, \ldots, m$. A total of 120 measurement matrices are generated, half of them are used for training and the other half are used for testing ($m = 60$ in this case).

In this setting, we still observe that the nuclear norm generally decreases over time, although not in a strict sense like before. We again observe that there are advantages in going from a single block to a multi-layer SNN from the generalization perspective.

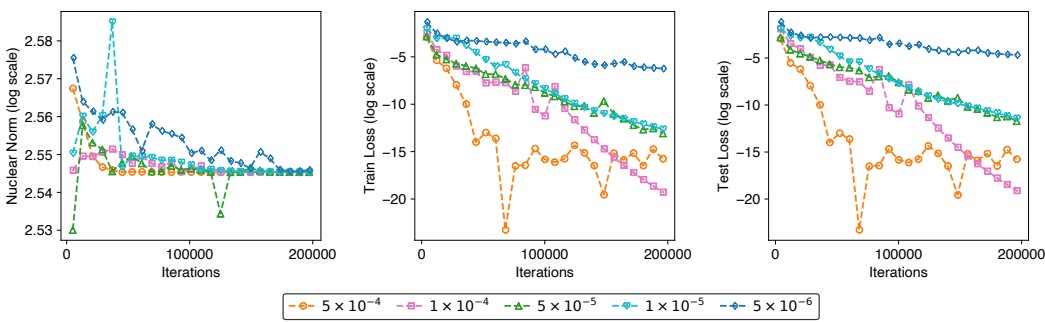

Figure 6: The evolution of SNN's nuclear norm, train error, and test error over time with learning rates (LR) of different magnitudes. From the left plot, the nuclear norm generally decreases over time and converges to the same value albeit with some erratic movements in the early iterations.

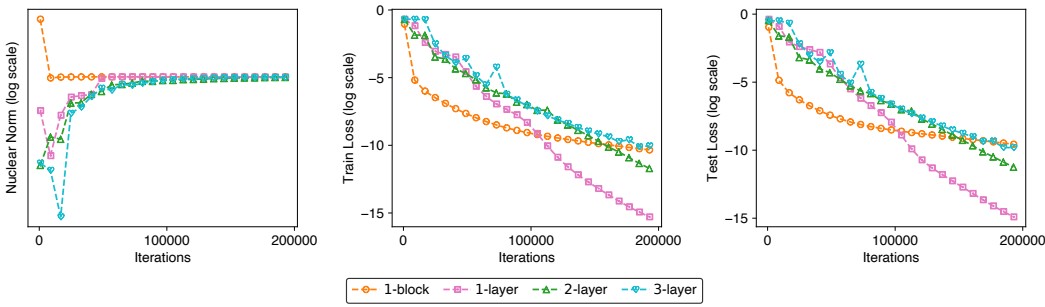

Figure 7: The evolution of SNN's nuclear norm, train error, and test error over time with various numbers of layers. We use a learning rate of $10^{-5}$. In the middle and right plots, it shows that the single-block 1-layer SNN is the easiest to optimize but does not result in the best generalization performance: the error decreases quickly at first, but then plateaus; on the other hand, for the more complex SNNs, the loss improves slowly but reaches better solutions.

## F    CODE AVAILABILITY

The python code used to conduct our experiments is publicly available at `https://github.com/porichoy-gupto/spectral-neural-nets`.

