# OpenReview forum: "Implicit regularization via Spectral Neural Networks and non-linear matrix sensing"
_ICLR.cc/2023/Conference — Submitted to ICLR 2023_

### Official Review · Reviewer_Ajzu · 2022-10-25

**Confidence:** 3
**Correctness:** 3
**Technical Novelty And Significance:** 4
**Empirical Novelty And Significance:** 2
**Recommendation:** 6

**Clarity, Quality, Novelty And Reproducibility:**

The paper has some merit in terms of the novel proposed architecture. But the presentation needs to be improved.

**Strength And Weaknesses:**

Strength:
1. The target problem is interesting.
2. This paper proposed a novel SNN architecture.
3. Both theoretical and numerical results are given to demonstrate the effectiveness of the proposed methods.

Weakness:
1. It would be better to present more details of the target problem and proposed architecture in the introduction and discuss a few more related works.
2. There are many parameters in SNN. It would be better to present more numerical experiments to study the sensitivity of SNN.
3. It would be interesting to perform SNN on some classical problems, such as matrix completion.


**Summary Of The Paper:**

This paper proposed a novel Spectral Neural Network for matrix learning problems. Some theoretical results are proven to show the effectiveness of SNN in matrix sensing. Several numerical experiments show the convergence of the matrix singular values.

**Summary Of The Review:**

This paper proposed a novel SNN architecture for matrix learning problems. More numerical experiments are required to show the effectiveness of SNN.

---

> ### Author Response · Authors · 2022-11-18
> **Presentation, Additional Experiments**
>
> We thank you for the positive comments and for acknowledging the novelty of our paper. We have responded to your comments about the paper's weaknesses below.
>
> #### **1. It would be better to present more details of the target problem and proposed architecture in the introduction and discuss a few more related works.**
>
> We have provided a brief discussion of the matrix sensing problem in the second paragraph and a high-level description of the proposed architecture in the final paragraph of the introduction. We have also discussed a few more related works. We want to emphasize that a key contribution of this work is to rigorously exhibit implicit regularization in the presence of non-trivial non-linearities, which is a novelty with regard to the existing literature. Of course, we would like to provide a more elaborate account in a fuller version of the paper; however, in the present version we are constrained by space limitations vis-a-vis a more detailed discussion of the general setup and the background.
>
> #### **2. There are many parameters in SNN. It would be better to present more numerical experiments to study the sensitivity of SNN.**
>
> We provide more numerical experiments in Appendix E. These experiments demonstrate the effect of implicit regularization in SNN with different learning rates (Figures 4 and 6) and different numbers of layers (Figures 5 and 7).
>
> In Figure 5, we can observe that there is a substantial difference in test error across different architectures. In this particular simulation, a 2-layer SNN achieves the lowest test error. This hints at the advantage of going beyond a single block to a multi-layer SNN architecture as we proposed in Section 3.
>
> We also have numerical experiments in which Assumption 1 (commuting measurement matrices) and Assumption 2 (spectral initialization) are relaxed. In these settings, we still observe that the nuclear norm generally decreases over time, although not in a strict sense like before. We again observe that there are advantages in going from a single block to a multi-layer SNN from the generalization perspective.
>
> #### **3. It would be interesting to perform SNN on some classical problems, such as matrix completion.**
>
> We agree with the reviewer that matrix completion would be an interesting problem. We want to highlight that matrix completion can be viewed as a special case of matrix sensing, in which the measurement matrices are elementary matrices in which a single entry equals $1$ and all other entries equal $0$.
>
> Without any further assumptions on the ground-truth matrix, the matrix completion problem is underdetermined since the missing entries could be assigned arbitrary values. Thus, we require some assumptions on the ground-truth matrix to create a well-posed problem. Common assumptions in the literature include low-rank, positive-definiteness, or maximal determinant.
>
> In our paper, we are interested in the implicit regularization effect of gradient descent. Our results suggest that running gradient descent on the matrix sensing problem leads to the minimum nuclear norm solution. We believe that it helps address the matrix completion problem in the sense that, in the setting of the ground-truth matrix being of low nuclear norm, SNN with gradient descent provably leads to a desirable solution.

---

### Official Review · Reviewer_xZxU · 2022-10-25

**Confidence:** 4
**Correctness:** 4
**Technical Novelty And Significance:** 3
**Empirical Novelty And Significance:** 2
**Recommendation:** 6

**Clarity, Quality, Novelty And Reproducibility:**

The paper is clear and I think the authors have done a good job in writing this work. Also, the spectral neural network architecture and the theoretical analysis all seem novel to me.

**Strength And Weaknesses:**

Strengths:

1) The paper proposes a novel neural network architecture (Spectral neural networks) for matrix learning problems.

2) The paper is well-written, and the theoretical discussion clearly states the assumptions and theorems.

Weaknesses:

1) The assumptions in section 4 look too strong to me. Especially the assumptions that the neural network and measurement matrices share the same left and right singular vectors seem restrictive to me. I recommend the authors include more explanation on the role of these assumptions in the theoretical analysis.

2) While I understand this is a theoretical work, the numerical results look preliminary.

**Summary Of The Paper:**

This paper proposes spectral neural networks (SNNs), an architecture that applies the nonlinear activation function to the singular values of a matrix, to demonstrate the role of implicit regularization in deep learning. Section 3 introduces the SNN architecture, and then Section 4 presents Theorem 3 suggesting that under the assumptions made the SNN training will converge to the nuclear-norm minimzing solution, indicating the implicit regularization of the gradient flow. A few numerical results are discussed in Section 5 to support SNNs and the implicit regularization in training them via gradient descent.

**Summary Of The Review:**

This paper proposes a novel neural network architecture for matrix learning problems. The paper contributes several nice ideas; however, it can still be improved by including additional numerical results and some explanation of the role and necessity of the assumptions in section 4.

---

> ### Author Response · Authors · 2022-11-18
> **Assumptions, Technical Contribution, Additional Experiments (Part 2)**
>
> #### **3. While I understand this is theoretical work, the numerical results look preliminary.**
>
> We provide additional numerical results in Appendix E, in which we investigate settings in which certain assumptions are relaxed. These experiments demonstrate the effect of implicit regularization in SNN with different learning rates (Figures 4 and 6) and different numbers of layers (Figures 5 and 7).
>
> In Figure 5, we can observe that there is a substantial difference in test error across different architectures. In this particular simulation, a 2-layer SNN achieves the lowest test error. This hints at the advantage of going beyond a single block to a multi-layer SNN architecture as we proposed in Section 3.
>
> We also have numerical experiments in which Assumption 1 (commuting measurement matrices) and Assumption 2 (spectral initialization) are relaxed. In these settings, we still observe that the nuclear norm generally decreases over time, although not in a strict sense like before. We again observe that there are advantages in going from a single block to a multi-layer SNN from the generalization perspective.
>
> ---
>
> [1] Li, Y., Ma, T., & Zhang, H. (2018). Algorithmic regularization in over-parameterized matrix sensing and neural networks with quadratic activations. In Conference On Learning Theory (pp. 2-47).
>
> [2] Gunasekar, S., Woodworth, B. E., Bhojanapalli, S., Neyshabur, B., & Srebro, N. (2017). Implicit regularization in matrix factorization. Advances in Neural Information Processing Systems, 30.
>
> [3] Arora, S., Cohen, N., Hu, W., & Luo, Y. (2019). Implicit regularization in deep matrix factorization. Advances in Neural Information Processing Systems, 32.

---

> ### Author Response · Authors · 2022-11-18
> **Assumptions, Technical Contribution, Additional Experiments (Part 1)**
>
> Thank you for your detailed comments and positive feedback. We have responded to your comments about the paper's weaknesses and your questions below.
>
> #### **1. The assumptions in section 4 look too strong to me. Especially the assumptions that the neural network and measurement matrices share the same left and right singular vectors seem restrictive to me.**
>
> We note that similar assumptions have been widely **adopted in the literature**. More specifically:
> 1. Our Assumption 1, commutative measurement matrices, is used by both Gunasekar et al. (2017, Theorem 1) [2] and Arora et al. (2019, Theorems 1 and 2) [3]. The only exception is Li et al. (2018) [1], in which instead of commutativity, the measurement matrices are required to satisfy a restricted isometry property.
> 2. Assumption 2 concerns the initialization procedure, which we have complete control over. Assumption 2, especially part (c) therein, only requires that the singular values of  $𝑋(0)$ ($X$ at initialization) are no greater than the singular values of $𝑋^*$ . Note that singular values of any matrix are non-negative real numbers. Therefore, this assumption can be satisfied by initializing the singular values of $𝑋$ close to $0$. Therefore, part (c) of Assumption 2 can be thought of as a much more general counterpart of the "near-zero" initialization that is widely adopted in the literature (Gunasekar et al. 2017 [2], Li et al. 2018 [1], Arora et al. 2019 [3]).
> 3. Assumption 3 imposes very mild restrictions on the non-linearity $\gamma$. Common non-linearities that are
> used in deep learning such as Sigmoid, ReLU (with truncation and quadratic smoothing), tanh, etc., satisfy the differentiability and non-decreasingness
> conditions, while boundedness can be achieved by truncating the outputs of these functions if
> necessary.
>
> In general, we argue that our assumptions are either prevalent in the literature (Assumption 1), or can be satisfied relatively easily (Assumptions 2 and 3). Therefore, as a first step towards analyzing the non-linear setting, it is natural for us to consider the same set of assumptions that are widely used in the literature. Relaxing Assumption 1 to accommodate for more general settings would be an interesting future direction.
>
> Over and above, we emphasize that a key contribution of this work is to rigorously exhibit implicit regularization in the presence of non-trivial non-linearities, which is a novelty with regard to the existing literature.
>
> #### **2. I recommend the authors include more explanation on the role of these assumptions in the theoretical analysis.**
>
> We thank the reviewer for the suggestions. The role of these assumptions is highlighted in the **short proof sketches** provided right below the theorems. For your convenience, we summarize them below:
> 1. Assumptions 1 and 2 together impose a distinctive structure on the gradient flow dynamics. We leverage the fact that the non-linearity $\gamma(\cdot)$ only changes the singular values while keeping the singular vectors intact. Therefore, the left and right singular vectors of $X(t)$ stay constant at $\Phi$ and $\Psi$ throughout the entire gradient flow dynamics. This allows us to derive closed-form expressions for the dynamics of the individual components of $X$.
> 2. Part (c) of Assumption 2 is used in Theorem 3 to ensure that the singular values of $X(0)$ at initialization are below the respective coordinates of $\sigma^\star$. We then use Assumption 3 to show that the derivatives of the singular values are positive, and hence the singular values can only increase over time. Together with the fact that the derivatives of the singular values at $\sigma^\star$ are exactly $0$, we show that the singular values of $X(t)$ converge to $\sigma^\star$ as $t \rightarrow \infty$.

---

### Official Review · Reviewer_p8b1 · 2022-11-01

**Confidence:** 4
**Correctness:** 2
**Technical Novelty And Significance:** 2
**Empirical Novelty And Significance:** 2
**Recommendation:** 3

**Clarity, Quality, Novelty And Reproducibility:**

The writing of this paper should be significantly improved. There are numerous inconsistencies, such as reused notations and lacking parenthesis for equation numbers. Sometimes equation is referred before its presence, such as Eq. 74. The code of SNN is provided for reproducibility.

**Strength And Weaknesses:**

Strength: this paper tries to use a nonlinear neural network to solve the matrix sensing problem.

Weakness: There are several major concerns and technical drawbacks of this paper.

1. Unclear motivation and contribution: previous works [Gunasekar et al. (2017) and Arora et al. (2019)] already showed that gradient flow by linear neural network achieves the minimum nuclear norm solution, this paper proposes a computationally expensive nonlinear neural network which leads to the same minimum nuclear norm solution (however, under very unrealistic assumption, see below). It is not clear what is the theoretical contribution or insight provided by this paper.

This paper presented a very vague description regarding the contribution, "Despite the large body of works studying implicit regularization, most of them consider the linear setting. It remains an open question to understand the behavior of gradient descent in the presence of
non-linearities..."

As stated above, one studies nonlinear neural network in the hope of brining theoretical benefits to the matrix sensing problem (or its implicit regularization property, that is, the minimum nuclear norm solution). However, this paper fails to provide any such benefit.

2. Unrealistic assumption: the main result of this paper relies on a key Assumption 2: note that one needs to access the information of the oracle (in terms of $\sigma^*$ obtained from $X^*$) in condition (c) of assumption 2, which is very unrealistic both theoretically and empirically.

3. Expensive computation: the feed-forward of the proposed network requires SVD on the input (see eq. (5)), and condition (a) in Assumption (2) is achieved by performing SVD of the sensing matrices, which are all computationally expensive, and in strong intransitive to existing works [Gunasekar et al. (2017), Arora et al. (2019), Li et al. (2018)].

**Summary Of The Paper:**

This paper proposes a nonlinear neural network (SNN) to solve the matrix sensing problem, and show that the gradient flow on SNN leads to a minimum nuclear norm solution to the matrix sensing problem.

**Summary Of The Review:**

While this paper proposes a nonlinear neural network to study the implicit regularization in matrix sensing problem, it suffers from multiple major technical drawbacks and unrealistic assumption. Due to the above major concerns, I recommend rejection.

---

> ### Author Response · Authors · 2022-11-18
> **Motivation, Contribution, and Assumptions (Part 2)**
>
> #### **2. Unrealistic assumption: the main result of this paper relies on a key Assumption 2: note that one needs to access the information of the oracle (in terms of $\sigma^\*$ obtained from $X^\*$) in condition (c) of assumption 2, which is very unrealistic both theoretically and empirically.**
>
> Assumption 2, especially part (c) therein, only requires that the singular values of $X(0)$ (X at initialization) are no greater than the singular values of $X^*$. Note that singular values of any matrix are non-negative real numbers. Therefore, this assumption can be satisfied by initializing the singular values of $X$ **close to $0$**.
>
> Therefore, part (c) of Assumption 2 can be thought of as a much more general counterpart of the "near-zero" initialization that is widely adopted in the literature (Gunasekar et al. 2017 [2], Li et al. 2018 [1], Arora et al. 2019 [3]).
>
> #### **3. Expensive computation: the feed-forward of the proposed network requires SVD on the input (see eq. (5)), and condition (a) in Assumption (2) is achieved by performing SVD of the sensing matrices, which are all computationally expensive, and in strong intransitive to existing works [Gunasekar et al. (2017), Arora et al. (2019), Li et al. (2018)].**
>
> While we agree that SNN is more computationally expensive than linear neural networks, we again emphasize that our contributions are mainly conceptual and theoretical in nature. In particular, we introduce the notion of SNNs in this work as a proof of concept. We believe that the many natural questions regarding them which emanate from this work, including questions of computational economy, will be addressed in subsequent research.
>
> ---
>
> [1] Li, Y., Ma, T., & Zhang, H. (2018). Algorithmic regularization in over-parameterized matrix sensing and neural networks with quadratic activations. In Conference On Learning Theory (pp. 2-47).
>
> [2] Gunasekar, S., Woodworth, B. E., Bhojanapalli, S., Neyshabur, B., & Srebro, N. (2017). Implicit regularization in matrix factorization. Advances in Neural Information Processing Systems, 30.
>
> [3] Arora, S., Cohen, N., Hu, W., & Luo, Y. (2019). Implicit regularization in deep matrix factorization. Advances in Neural Information Processing Systems, 32.

---

> ### Author Response · Authors · 2022-11-18
> **Motivation, Contribution, and Assumptions (Part 1)**
>
> Thank you for your comments and suggestions. We have responded to your comments below.
>
> #### **1. Unclear motivation and contribution: previous works [Gunasekar et al. (2017) and Arora et al. (2019)] already showed that gradient flow by linear neural network achieves the minimum nuclear norm solution, this paper proposes a computationally expensive nonlinear neural network which leads to the same minimum nuclear norm solution (however, under very unrealistic assumption, see below). It is not clear what is the theoretical contribution or insight provided by this paper.**
>
> The pursuit of deep learning theory as a field is to explain the generalization ability of deep neural networks, and more broadly, to achieve a theoretical understanding of how and why deep learning works --- in other words, to explain the behaviour of deep neural nets which have been largely successful in many empirical applications. Non-linearities acting on each layer are fundamental features of neural networks, and any theory attempting to explain their behaviour needs to address the foundational aspect of non-linearities. Without these non-linearities, a network of any depth is just a linear function.
>
> Implicit regularization (IR) properties of the gradient descent algorithm have been observed in various neural networks employed in practice. Such networks are of course highly non-linear. Explaining IR in a non-linear setting is quite challenging from a mathematical perspective. Indeed, existing works that provide theoretical explanations for IR mainly deal with linear neural networks (except for Li et al. (2018) [1], who also allow an algebraically specific type of quadratic nonlinearity).
>
> In this work, our principal aim is to bridge this gap, and put forward a theoretical analysis of implicit regularization in neural networks in the presence of non-linearities. The proposed SNN architecture provides a highly non-linear set-up, while at the same time being amenable to rigorous mathematical analysis, thanks to its gradient updates taking place in spectral coordinates.
>
> We emphasize that our main contribution is the theoretical demonstration of IR in a highly non-linear setting, which as discussed above has been lacking in the existing literature.
>
> The SNN architecture, while computationally more expensive than linear neural networks, is potentially more expressive and suitable for more complex matrix sensing tasks. We leave the study of these aspects to future work.
>
> To provide a more detailed picture, our paper contributes to the progress of deep learning theory as follows:
>
> 1. We propose the concept of spectral non-linearities, which promise to be of interest both from a theoretical and practical point of view. To wit, the non-linear activation function is applied on the singular values rather than entry-wise. **Theorem 1** provides closed-form expressions for the dynamics of the individual components of $X$. We want to highlight that the compact analytical expression and the simplicity of the gradient flow dynamics on the components are a direct result of the spectral non-linearity. Notice that our results encompass a more general class of models, on which linear neural networks are a special case and can be recovered by setting the non-linear function to be the identity function, i.e., $\gamma(x)=x$.
> The notion of spectral non-linearities, as well as the analysis of gradient flow in Theorem 1, contains many **novel** and non-standard arguments, which differ substantially from the analyses in existing works. We believe that these novel ideas will be of interest to the community, especially in the theoretical aspects of deep learning.
> 2. The limiting matrix output by the network is the best approximation of $X^∗$ among matrices with (the columns of) $\Phi$ and $\Psi$ as their left and right singular vectors in **Theorem 2**. Thus, we show that the final output of the SNN is, conceptually speaking, the best possible in a reasonable class within the ambit of the setup. We also show that convergence to the limiting matrix happens at an exponentially fast rate.
> 3. Finally, we rigorously exhibit the phenomenon of **implicit regularization** in our model. To this end, **Theorem 3** shows that the limiting matrix obtained from the gradient flow dynamics will attain the minimum nuclear norm. Our results are in spirit of similar flavour to those in Gunasekar et al. (2017) [2] and Arora et al. (2019) [3], but the crucial point is that our approach addresses **a more general class of neural networks** with the presence of non-linearities, and within this class, the linear neural networks of the existing literature constitute special cases.

---

### Official Review · Reviewer_yAZt · 2022-11-04

**Confidence:** 3
**Correctness:** 3
**Technical Novelty And Significance:** 3
**Empirical Novelty And Significance:** 4
**Recommendation:** 8

**Clarity, Quality, Novelty And Reproducibility:**

Clarity - The paper is well-written and clear. Quality - This paper has good quality of both the writing and the solution. Originality - Good.

**Strength And Weaknesses:**

This paper focuses on an interesting topic of the implicit regularization for neural network with non-linear activation functions. It should attract a lot of attention from the community. The paper is well-written and well-organized, every concept is well explained and easy to follow. I feel really pleasure reading the paper. To analyze the implication regularization for matrix sensing problem, the authors proposed the spectral neural network architecture. Such structure incorporate the singular values and singular vectors of the matrix, instead of the entries, which is quite interesting. Besides, the following theoretical analysis and empirical evaluation support the claims and shows the effectiveness of the proposed solution.

Overall, this paper is good. I have some minor suggestions and questions.
First, for the matrix sensing problem this paper focuses on, is there any assumption on the matrix X*, e.g., low rank matrix? I didn't see any claims in the problem setting section (correct me if I am wrong).
The illustration of the SNN structure is a little bit confusing. For example, is every blue rectangular in the SNN corresponding to the block, or it is just one row of the block? I would suggest the authors to provide more details of the figure to help readers better understand the structure. Besides, what is the relation between the K in the block and the L in the SNN structure? How to choose the value of K and L in experiments, are they related to the property of the matrix?
Minor: the size of the font could be enlarged in the result figures, e.g. ,Figure 3

**Summary Of The Paper:**

This paper focuses on the implicit regularization phenomenon in neural network with gradient descent. Different from the previous work, the authors works on the more general neural nets with non-linear activation functions. In particular, the authors demonstrated the implicit regularization phenomenon with non-linear activation function for matrix sensing problem. A new neural network structure called spectral neural network is proposed for solving the matrix learning problem. Both theoretical analysis and empirical evaluations are reported to support the proposed solution.

**Summary Of The Review:**

This paper focuses on a really interesting topic and proposed the first analysis of the implicit regularization for neural network with non-linear activation function (for matrix sensing problem). The proposed SNN structure is well suited for the matrix sensing problem and the corresponding theoretical analysis supports the claims. Empirical evaluations also demonstrates the effectiveness of the proposed solution. Overall, this paper has high quality and I would recommend accept the paper.

---

> ### Author Response · Authors · 2022-11-18
> **Assumptions, Clarifications on SNN Structure**
>
> Thank you for your encouraging comments and suggestions. We are genuinely appreciative of your effort to thoroughly understand our paper. We have responded to each of your concerns below.
>
> #### **1. First, for the matrix sensing problem this paper focuses on, is there any assumption on the matrix X\*, e.g., low-rank matrix?**
>
> You are correct that we did not impose any assumptions on the ground-truth matrix $X^*$. We only have assumptions on the measurement matrices (assumption 1), the matrix at initialization $X(0)$ (assumption 2), and the non-linearity (assumption 3).
>
> #### **2. The illustration of the SNN structure is a little bit confusing. For example, is every blue rectangular in the SNN corresponding to the block, or it is just one row of the block? I would suggest the authors to provide more details about the figure to help readers better understand the structure.**
>
> We thank the reviewer for the suggestions. We have revised the Figure 1 caption to include a more detailed description of the SNN architecture.
>
> In Figure 1, the blue rectangular regions in the SNN correspond to the entire blocks, and not just one row. Notice that each block takes as input $K$ matrices, and outputs one matrix, both the input and output matrices are of size $\mathbb{R}^{d_1 \times d_2}$ (which is of the same size as $X^*$).
>
> #### **3. Besides, what is the relation between the K in the block and the L in the SNN structure? How to choose the value of K and L in experiments, are they related to the property of the matrix?**
>
> $K$ represents the number of input matrices, where as $L_i$ represents the number of neurons in layer $i$.
>
> In the SNN architecture, the number of input matrices to a block equals the number of neurons in the previous layer. For example, blocks in layer 1 have $K=L_0$, blocks in layer 2 have $K=L_1$, etc. Generally, blocks in layer $i$ have $K=L_{i-1}$.
>
> #### **4. How to choose the value of K and L in experiments, are they related to the property of the matrix?**
>
> The problem of choosing $K$ and $L_i$'s is akin to the problem of choosing the number of layers and hidden neurons in a neural network. They are hyper-parameters in the SNN model, and thus we can use a wide range of hyper-parameter tuning techniques available in deep learning.
>
> The $K$ and $L_i$'s in the experiments are chosen by trial and error. We, therefore, believe that one may get improved performance using more sophisticated hyper-parameter optimization techniques.

---

### Author Response · Authors · 2022-11-18
**General Response to All Reviewers**

We thank all reviewers for their time and efforts in evaluating our paper and for their detailed comments and suggestions. We hope our responses and answer to your questions will alleviate your concerns and further improve your opinion of our work. We have also uploaded a revised version of the paper. If you have additional questions, we would be happy to address them.

---

### Decision · Program_Chairs · 2023-01-20

**Decision:**

Reject

**Justification For Why Not Higher Score:**

Although the model seems interesting the assumptions are too strong, which makes the result less interesting. Especially considering that the regularization effect is still minimum nuclear norm, which is not even true for linear settings without Assumption 1.

**Justification For Why Not Lower Score:**

N/A

**Metareview: Summary, Strengths And Weaknesses:**

This paper analyzes a model of non-linear matrix sensing and prove implicit regularization effects for gradient flow. Under some assumptions and a nonlinear model, the paper shows that gradient flow converges to the minimum nuclear norm solution. The nonlinear architecture seems to be of interest independently. The reviewers agree that the analysis of nonlinear models in matrix sensing is novel and interesting. However there are many concerns about the strength of assumptions. For example, Assumption 1 was only used in earlier implicit regularization works and recent results have actually shown (in linear setting) that the dynamics of gradient flow does not give the minimum nuclear norm solution; Assumption 2 uses a spectral initialization which depends on the sensing matrix, and that is a bit strange as it's not clear whether the final solution is caused by this careful initialization or implicit regularization.

**Summary Of Ac-Reviewer Meeting:**

N/A